# Evidence linking APOBEC3B genesis and evolution of innate immune antagonism by gamma-herpesvirus ribonucleotide reductases

Sofia N Moraes[1], Jordan T Becker[1], Seyed Arad Moghadasi[1], Nadine M Shaban[1], Ashley A Auerbach[1,2], Adam Z Cheng[1], Reuben S Harris[1,2,3]*

[1]Department of Biochemistry, Molecular Biology, and Biophysics, Institute for Molecular Virology, Masonic Cancer Center, University of Minnesota, Minneapolis, United States; [2]Department of Biochemistry and Structural Biology, University of Texas Health San Antonio, San Antonio, United States; [3]Howard Hughes Medical Institute, University of Texas Health San Antonio, San Antonio, United States

**Abstract** Viruses have evolved diverse mechanisms to antagonize host immunity such as direct inhibition and relocalization of cellular APOBEC3B (A3B) by the ribonucleotide reductase (RNR) of Epstein-Barr virus. Here, we investigate the mechanistic conservation and evolutionary origin of this innate immune counteraction strategy. First, we find that human gamma-herpesvirus RNRs engage A3B via largely distinct surfaces. Second, we show that RNR-mediated enzymatic inhibition and relocalization of A3B depend upon binding to different regions of the catalytic domain. Third, we show that the capability of viral RNRs to antagonize A3B is conserved among gamma-herpesviruses that infect humans and Old World monkeys that encode this enzyme but absent in homologous viruses that infect New World monkeys that naturally lack the *A3B* gene. Finally, we reconstruct the ancestral primate A3B protein and demonstrate that it is active and similarly engaged by the RNRs from viruses that infect humans and Old World monkeys but not by the RNRs from viruses that infect New World monkeys. These results combine to indicate that the birth of *A3B* at a critical branchpoint in primate evolution may have been a driving force in selecting for an ancestral gamma-herpesvirus with an expanded RNR functionality through counteraction of this antiviral enzyme.

*For correspondence:
rsh@uthscsa.edu

## Editor's evaluation

This important work builds on the conceptual framework of host-pathogen interactions and co-evolution, adding new examples of co-divergence of primate herpesviruses with their respective host restriction factors. The authors convincingly outline the degree to which their initial findings (BORF2 and A3B interactions) are conserved across other herpesvirus RNRs, and place them in the context of the evolution of the A3 gene locus and expansion. This work will be of great interest to virologists. Especially those that work in the field of host pathogen evolution and the molecular arms race.

## Introduction

Cell-intrinsic innate immune proteins, or 'restriction factors', contribute to the first line of defenses against incoming viruses (reviewed by *Duggal and Emerman, 2012*). A prototypical example is the APOBEC3 (A3) family of single-stranded (ss)DNA cytosine deaminases, which provide broad and overlapping protection against viral infections (reviewed by *Harris and Dudley, 2015*; *Simon*

*et al., 2015*). Classically, A3s exert antiviral activity by catalyzing cytosine deamination in exposed ssDNA replication intermediates, often resulting in lethal mutagenesis of viral genomes. Humans encode seven A3 enzymes (A3A-D, A3F-H), which have been implicated in the restriction of retroviruses (HIV and HTLV) (*Sheehy et al., 2002*; *Harris et al., 2003*; *Wiegand et al., 2004*; *Sasada et al., 2005*; *Dang et al., 2006*), human endogenous retroviruses (HERV) (*Esnault et al., 2005*; *Lee et al., 2008*), human hepadnaviruses (HBV) (*Turelli et al., 2004*; *Suspène et al., 2005*), human papillomaviruses (HPV) (*Vartanian et al., 2008*), polyomaviruses (BK) (*Peretti et al., 2018*), and most recently the gamma-herpesvirus Epstein-Barr virus (EBV; *Cheng et al., 2019b*). It is difficult to predict which subset of A3 enzymes has the potential to restrict a given virus, but differences in the subcellular localization properties of these enzymes provide useful insights into their antiviral targets. For instance, cytoplasmic A3s like A3G and A3F potently restrict HIV-1 by binding to viral genomic RNAs in the cytoplasm, packaging into budding virions at the cell membrane, and deaminating viral ssDNA intermediates during reverse transcription (*Harris and Liddament, 2004*; *Mangeat et al., 2003*; *Navarro et al., 2005*). It thus follows that A3B—the only constitutively nuclear human A3 enzyme—may pose a threat to the genomic integrity of viruses that undergo replication in the nucleus, such as EBV and related herpesviruses.

Viruses that are susceptible to restriction by cellular A3 enzymes have evolved equally potent counteraction mechanisms to ensure successful replication (reviewed by *Harris and Dudley, 2015*; *Simon et al., 2015*; *Malim and Bieniasz, 2012*). The most well-studied A3 antagonist is the lentiviral accessory protein Vif, which nucleates the formation of an E3 ubiquitin ligase complex to promote the selective degradation of up to five different cytoplasmic A3 enzymes: A3C, A3D, A3F, A3G, and A3H (reviewed by *Harris and Dudley, 2015*; *Hu et al., 2021*; *Uriu et al., 2021a*). Extensive mutagenesis analyses revealed that HIV-1 Vif evolved to use distinct binding surfaces in order to recognize different A3s, as underscored by separation-of-function mutants of Vif that selectively degrade only a subset of these enzymes (reviewed by *Hu et al., 2021*; *Kitamura et al., 2011*; *Aydin et al., 2014*). These studies have been essential to our understanding of how viruses adapt to overcome innate host defenses and, collectively, highlight the importance of elucidating structure-function relationships between proteins at the host-pathogen interface.

HIV-1 and related retroviruses are thought to be primary targets of A3 enzymes due to obligate ssDNA replication intermediates, abundant evidence for A3-catalyzed hypermutation of viral genomes, and a potent A3 counteraction mechanism. Reports of DNA viruses exhibiting signatures of A3-mediated mutagenesis suggest that these viruses may also be susceptible to restriction by A3 enzymes (*Harris and Dudley, 2015*; *Shapiro et al., 2021*), although the identification of viral antagonists has been challenging. We recently discovered that the large double-stranded (ds)DNA herpesvirus EBV uses a two-pronged approach to counteract the antiviral activity of A3B. The large subunit of the EBV ribonucleotide reductase (RNR), BORF2, directly inhibits A3B catalytic activity and relocalizes this enzyme from the nucleus to cytoplasmic aggregates—most likely to protect viral genomes from deamination during lytic replication (*Cheng et al., 2019b*). Consistent with this interpretation, BORF2-null EBV accumulates A3B signature C/G to T/A mutations upon lytic reactivation and exhibits significantly lower infectivity (*Cheng et al., 2019b*).

The deamination activity of all A3 family enzymes requires a zinc-coordinating motif, which forms the core of the catalytic site and is surrounded by three loop regions (L1, L3, and L7) that contribute to binding of ssDNA substrates (*Shi et al., 2017*; *Kouno et al., 2017*; *Maiti et al., 2018*). These loops vary in size and amino acid composition and account for differences in A3 catalytic rates and local target sequence preferences (reviewed by *Silvas and Schiffer, 2019*; *Maiti et al., 2021*). The neutralization of A3B by EBV BORF2 is the first example of direct inhibition of A3 ssDNA deaminase activity by another protein, and our recently published cryo-EM structure of EBV BORF2 in complex with the catalytic domain of A3B (A3Bctd) provides critical insights into the structural basis and molecular mechanism of this unanticipated host-pathogen interaction (*Shaban et al., 2022*). Specifically, our cryo-EM studies revealed that EBV BORF2 binds to the L1 and L7 regions of A3Bctd, resulting in a high-affinity interaction that effectively blocks the A3B active site from binding to ssDNA substrates and catalyzing deamination. Recently, we and others have shown that, in addition to EBV BORF2, the large RNR subunit from related human herpesviruses can also bind and relocalize A3B—as well as the highly similar A3A enzyme—suggesting that this A3 counteraction mechanism may be conserved evolutionarily (*Stewart et al., 2019*; *Cheng et al., 2019a*; *Cheng et al., 2021*).

Here, we investigate the molecular and functional conservation of A3 antagonism by the RNR subunits of different herpesviruses using structurally and evolutionarily informed approaches. We find that the RNR subunits of EBV and the related human gamma-herpesvirus Kaposi's sarcoma-associated herpesvirus (KSHV) interact with A3B and A3A by binding to distinct loop regions near the catalytic site, which accounts for differences in binding selectivity and capacity to inhibit A3 enzymatic activity. We also uncover potential coevolutionary interactions between A3 enzymes and primate gamma-herpesviruses by discovering a link between the generation of the *A3B* gene in primates and the evolution of A3-neutralization functionalities in viral RNRs. Last, we reconstruct an ancestral primate A3B enzyme and show that it is bound, relocalized, and inhibited by the RNRs of gamma-herpesviruses that infect primate species that encode *A3B* but not by the RNRs from primate viruses that infect species that naturally lack this gene. Taken together, our data provide evidence of a conserved and likely ancient A3 antagonism mechanism in primate gamma-herpesviruses and offer insights into how genetic conflicts with host restriction factors may have affected—and likely continue to affect—the structure and function of viral proteins such as the herpesvirus RNR.

## Results

### Structural differences and similarities between EBV and KSHV RNR large subunits

Our recent cryo-EM studies of EBV BORF2-A3B complexes revealed a large binding surface composed of multiple structural elements from each protein (*Shaban et al., 2022*). Specifically, loops 1 (L1) and 7 (L7) of the A3B catalytic domain (A3Bctd) are sequestered by multiple interactions with BORF2 including contacts with residues L133 and Y134 of a novel short helix insertion (SHI) and residues R484 and Y481 of an additional helical loop structure (HLS). Y134 additionally forms a network of stabilizing interactions with other BORF2 residues including Q476 and R484 (*Shaban et al., 2022*; depicted in *Figure 1A*). The BORF2 homolog from related gamma-herpesvirus KSHV (ORF61) has also been shown to bind A3B (*Cheng et al., 2019b*; *Cheng et al., 2019a*). However, an amino acid sequence alignment of EBV BORF2 and KSHV ORF61 indicates limited identity in the SHI and HLS regions (*Figure 1B*; complete sequence alignment in *Figure 1—figure supplement 1A*). We therefore used RoseTTAFold (*Baek et al., 2021*) to generate a structural model of KSHV ORF61 for three-dimensional comparisons (*Figure 1C*). Although sharing only ~42% primary amino acid identity, the predicted model of KSHV ORF61 overlays well with the cryo-EM structure of EBV BORF2 with root mean square deviation of 1.26 Å. Interestingly, in addition to similar HLS regions, EBV BORF2 and the predicted structure of KSHV ORF61 share a similar sized SHI region (depicted in red in *Figure 1C*), which is lacking in nonviral class Ia RNRs such as those from *Escherichia coli* and humans (*Shaban et al., 2022*).

To gain further insights into the potential role of KSHV ORF61 SHI and HLS regions in binding to A3B, we used in silico docking to investigate how ORF61 and A3Bctd might interact. RoseTTAFold models of KSHV ORF61 and A3Bctd were aligned to the cryo-EM BORF2-A3Bctd complex using PyMOL (*DeLano, 2002*) to create an initial docking pose followed by local protein-protein docking using RosettaDock 4.0 (*Lyskov et al., 2013*; *Marze et al., 2018*). Models were ranked by Critical Assessment of Predicted Interactions (CAPRI) score and the best structure was selected based on overall interface energy score (Materials and methods and *Figure 1—figure supplement 1B*). Several high-scoring models suggested that the ORF61 SHI region may interact with the loop 3 (L3) region of A3B, instead of L1 and L7 as for BORF2 (*Figure 1D–E*). Additionally, the predicted HLS region of KSHV ORF61 appears to be positioned further away from the RNR-A3Bctd binding interface and may not contribute to the interaction (*Figure 1E*). Taken together, these in silico predictions suggest that EBV BORF2 and KSHV ORF61 may differentially engage the catalytic domain of A3B by interacting with distinct loop regions. This possibility is tested in the following sections.

### Involvement of the A3B and A3A loop 7 regions in binding to gamma-herpesvirus RNRs

Recent biochemical and structural studies demonstrated that direct contacts between EBV BORF2 and the L7 region of A3Bctd are essential for this host-pathogen interaction (*Cheng et al., 2019b*; *Shaban et al., 2022*). A3Bctd is closely related to A3A (>90% amino acid identity including identical L7 regions; *Figure 2A*), suggesting that a similar interaction with L7 could explain EBV BORF2

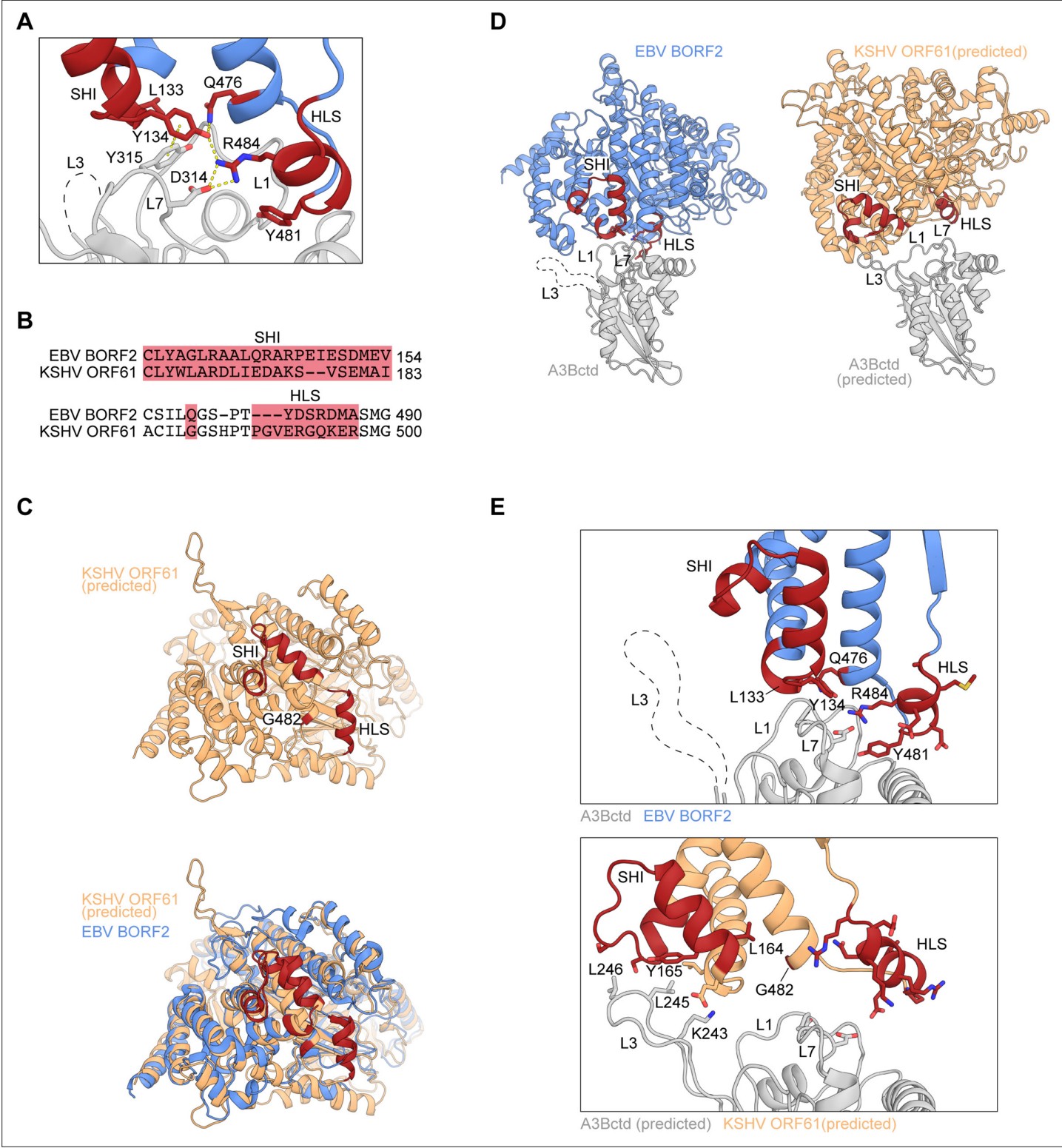

**Figure 1.** Structural differences and similarities between Epstein-Barr virus (EBV) and Kaposi's sarcoma-associated herpesvirus (KSHV) ribonucleotide reductase (RNR) large subunits. (**A**) Zoom-in of the EBV BORF2-A3Bctd interface (pdb 7rw6, chains A and B) showing a network of interactions between BORF2 (blue) and A3Bctd L7 and L1 regions (gray). BORF2 short helix insertion (SHI), helical loop structure (HLS), and interacting residue Q476 are shown in red. Gray dashed line represents the L3 region not visible by cryo-EM. (**B**) Amino acid sequence alignment of the SHI and HLS regions of EBV BORF2 and KSHV ORF61 (complete alignment in *Figure 1—figure supplement 1A*). (**C**) RoseTTAFold-predicted structure of KSHV ORF61 (orange, above) overlayed with EBV BORF2 (pdb 7rw6, chain A; blue, below). HLS and SHI regions in both proteins and residue G482 in ORF61 are shown in red.

*Figure 1 continued on next page*

*Figure 1 continued*

(**D–E**) Cryo-EM structure of EBV BORF2-A3Bctd (pdb 7rw6, chains A and B) and highest-ranked protein-protein docking model of KSHV ORF61-A3Bctd generated using RosettaDock (***Figure 1—figure supplement 1B***). Panel D shows the full complexes with key regions labeled. Panel E shows a zoom-in of the RNR-A3ctd interface (BORF2, blue; ORF61, orange; A3Bctd gray; SHI and HLS regions, red). A3Bctd loops are labeled L1, L3, and L7 (gray dashed line represents the L3 region not visible by cryo-EM).

The online version of this article includes the following figure supplement(s) for figure 1:

**Figure supplement 1.** EBV BORF2 and KSHV ORF61 amino acid sequence alignment and summary of protein-protein docking models generated.

ability to engage A3B and A3A, but not other A3 family members. Recent work also revealed that KSHV ORF61 is similarly capable of binding and relocalizing A3B and A3A, but not other A3 enzymes (***Cheng et al., 2019a***).

To test whether binding to L7 is a conserved requirement for viral RNR-A3 interaction, we generated a series of chimeric A3 proteins with reciprocal L7 swaps and tested their ability to bind to EBV BORF2 and KSHV ORF61 in co-immunoprecipitation (co-IP) experiments (***Figure 2B***). 293T cells were transfected with FLAG-tagged viral RNRs and EGFP-tagged A3s, subjected to anti-FLAG affinity purification, and analyzed by immunoblotting. As expected from our prior studies, replacing L7 of A3Bctd or A3A with L7 of A3Gctd completely abolishes interaction with EBV BORF2 (***Figure 2B***, left panel, lanes 1–4) and, reciprocally, replacing L7 of A3Gctd with L7 of A3A/Bctd endows binding (***Figure 2B***, left panel, lanes 5–6). However, we were surprised to find that replacing L7 of A3Bctd or A3A with L7 of A3Gctd does not affect binding by KSHV ORF61 (***Figure 2B***, right panel, lanes 1–4). Accordingly, replacing L7 of A3Gctd with L7 of A3A/Bctd does not enable pull-down by KSHV ORF61 (***Figure 2B***, right panel, lanes 5–6).

An additional defining feature of the interaction between A3B/A and viral RNRs is relocalization from the nucleus to the cytoplasm (***Cheng et al., 2019b***; ***Stewart et al., 2019***; ***Cheng et al., 2019a***). To evaluate the subcellular localization phenotypes of the L7 chimeras described above, we performed quantitative fluorescence microscopy experiments using HeLa cells co-transfected with EGFP-tagged A3s and FLAG-tagged viral RNRs (***Figure 2C***, representative images). In agreement with our co-IP results, replacing L7 of A3Bctd or A3A with L7 of A3Gctd markedly decreases EBV BORF2-mediated relocalization, although low levels of A3Bctd L7G relocalization can still be detected in a subset of cells. Reciprocally, replacing L7 of A3Gctd with L7 of A3A/Bctd enables effective relocalization (***Figure 2C***, representative images; ***Figure 2D–E***, quantification). In further agreement with our co-IP results, swapping L7 does not decrease KSHV ORF61-mediated relocalization of A3Bctd or A3A, nor does it increase A3Gctd relocalization (***Figure 2C***, representative images; ***Figure 2D–E***, quantification). Collectively, our experimental data support the in silico protein docking predictions and indicate that, despite an overall high degree of predicted structural similarity, the RNR subunits from EBV and KSHV engage A3Bctd and A3A through at least partially distinct surfaces.

## Role of A3B and A3A loop 3 in binding to gamma-herpesvirus RNRs

L3 is one of three loops that surround the A3 active site (***Figure 2A***). In our recent cryo-EM studies of the EBV BORF2-A3Bctd complex, we were unable to model A3Bctd L3 residues due to the flexibility of this region (***Shaban et al., 2022***; dashed region in ***Figure 3A***). In silico protein-protein docking predictions in ***Figure 1D–E*** suggest that this loop may be important for KSHV ORF61 binding. Therefore, to examine whether L3 participates in viral RNR-A3 interactions, we performed co-IP experiments using a panel of chimeric A3 proteins with reciprocal L3 swaps (***Figure 3B***).

Once again, we observed markedly distinct phenotypes in co-IPs with EBV BORF2 and KSHV ORF61. For EBV BORF2, similar pull-down levels are observed for wild-type (WT) versus L3-swapped A3s (***Figure 3B***, left panel, lanes 1–4), whereas replacing L3 of A3Bctd or A3A with L3 of A3Gctd nearly abolishes the interaction with KSHV ORF61 (***Figure 3B***, right panel, lanes 1–4). Additionally, replacing L3 of A3Gctd with L3 of A3A/Bctd is sufficient to promote modest immunoprecipitation with KSHV ORF61 (***Figure 3B***, right panel, lanes 5–6). These results indicate that the L3 region is critical for KSHV ORF61 binding and likely to have little or no role in binding to EBV BORF2.

To evaluate the subcellular localization phenotypes of the L3 chimeras described above, we performed quantitative fluorescent microscopy experiments using HeLa cells (***Figure 3C***, representative images). Similar to our L7 results, microscopy observations are largely concordant with co-IP phenotypes. For EBV BORF2, L3 swaps have no substantial effect on A3 relocalization, whereas

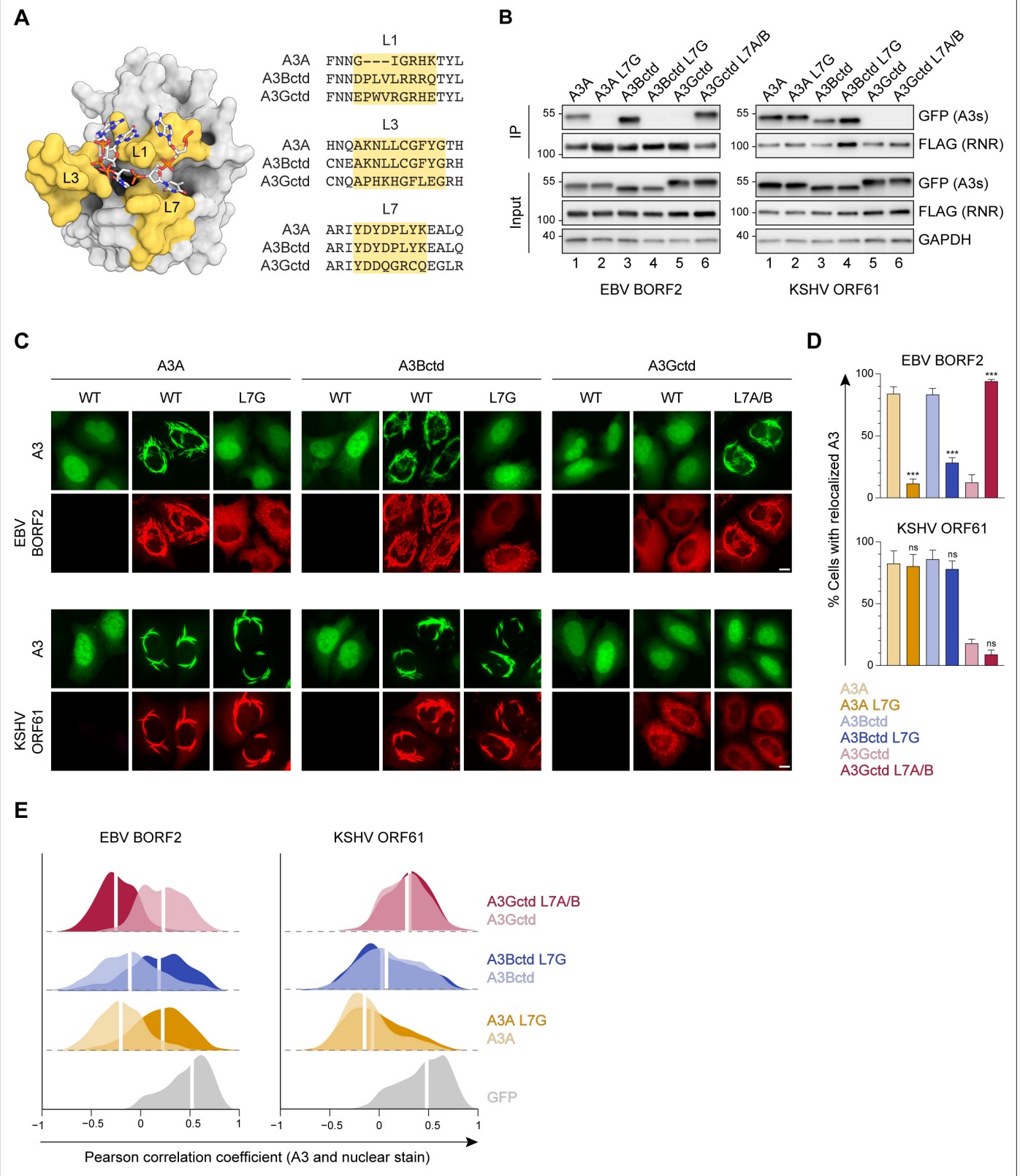

**Figure 2.** Involvement of the A3B and A3A loop 7 regions in binding to gamma-herpesvirus ribonucleotide reductases (RNRs). (**A**) Surface-filled representation of single-stranded DNA (ssDNA) bound by an A3 (pdb 5sww, chains A and E). L1, L3, and L7 regions are colored in yellow with amino acid alignments to the right. (**B**) Co-immunoprecipitation (Co-IP) of A3 L7 chimeras with Epstein-Barr virus (EBV) BORF2 and Kaposi's sarcoma-associated herpesvirus (KSHV) ORF61. FLAG-tagged RNR subunits were co-expressed with the indicated A3-EGFP constructs in 293T cells, affinity purified, and

*Figure 2 continued on next page*

*Figure 2 continued*

analyzed by immunoblotting to detect co-purifying A3 proteins. (**C**) Representative IF microscopy images of HeLa cells co-transfected with FLAG-tagged EBV BORF2 or KSHV ORF61 (red) and the indicated A3-EGFP constructs (green). Scale = 10 μm. (**D–E**) Quantification of A3-EGFP relocalization in HeLa cells expressing EBV BORF2 or KSHV ORF61. Panel D shows the percentage of cells exhibiting relocalized A3-EGFP (mean ± standard deviation, n≥100 cells per condition). Statistical analyses were performed using unpaired t-tests to determine significant changes in the relocalization of L7 swapped A3s (darker shade) relative to wild-type (WT) (lighter shade; n=3 independent experimental replicates; ns, not significant p>0.5; *p≤0.5; **p≤0.01; ***p≤0.001). Panel E shows Pearson correlation coefficient (PCC) values for A3-EGFP (or EGFP alone as a control) and Hoechst (nuclear stain). Density curves show the PCC value distribution in each condition, and the white vertical lines indicate the median PCC value (n≥100 cells per condition).

The online version of this article includes the following source data for figure 2:

**Source data 1.** File contains original immunoblots for *Figure 2B*.

replacing L3 of A3Bctd or A3A with the corresponding region of A3Gctd causes a marked decrease in relocalization by KSHV ORF61 (*Figure 3C*, representative images; *Figure 3D–E*, quantification). Notably, KSHV ORF61 is able to relocalize the chimeric A3Gctd L3A/Bctd protein, albeit to considerably lesser extents compared to WT A3Bctd or A3A. These data provide further support for the in silico predictions above by demonstrating the importance of the L3 region for binding to KSHV ORF61 and promoting cytoplasmic aggregate formation in living cells.

## Role of A3B and A3A loop 1 in binding to gamma-herpesvirus RNRs

An unanticipated discovery made possible by our recent EBV BORF2-A3Bctd cryo-EM structure is the existence of an extensive interaction between the short HLS region of BORF2 and L1 of A3Bctd (*Shaban et al., 2022*). This L1 interaction accounts for EBV BORF2 binding preferentially to A3B as opposed to A3A. We therefore compared the role of the L1 region of A3Bctd versus A3A in the interaction of these enzymes with EBV BORF2 and KSHV ORF61. First, we confirmed our recent report by showing that replacing L1 of A3A with L1 of A3Bctd leads to a marked increase in EBV BORF2 co-IP (*Figure 4A*, left panel, lanes 1–2). A strong increase in pull-down is also visible when L1 of A3A is replaced with L1 of A3Gctd, possibly due to the similarities between the L1 regions of A3Bctd and A3Gctd, where each enzyme has a 3-residue motif that is absent in A3A (PLV and PWV, respectively; *Figure 2A*). Reciprocally, replacing L1 of A3Bctd with L1 of A3A—but not with L1 of A3Gctd—dramatically diminishes the interaction with EBV BORF2 (*Figure 4A*, left panel, lanes 4–5).

In contrast, KSHV ORF61 does not appear to bind more strongly to A3Bctd compared to A3A (*Figure 4A*, right panel, compare lanes 1 and 4). Moreover, L1 swaps between A3Bctd and A3A have no effect on KSHV ORF61 co-IPs (*Figure 4A*, right panel, lanes 1, 2, 4, and 5). However, interestingly, replacing L1 of either A3Bctd or A3A with residues from A3Gctd decreases interaction with KSHV ORF61 (*Figure 4A*, right panel, lanes 3 and 6). This may be attributable to the presence of a unique bulky tryptophan residue in A3Gctd L1 potentially clashing with a nearby interacting surface (*Figure 1* and *Figure 2A*).

We next assessed the subcellular localization phenotypes of these L1 chimeras using quantitative fluorescence microscopy (*Figure 4B*, representative images; *Figure 4C–D*, quantification). In contrast to the co-IP results above, swapping the L1 regions of A3Bctd and A3A does not overtly affect relocalization by EBV BORF2. This agrees with our published results (*Shaban et al., 2022*) and supports a model in which compensatory avidity interactions during cytoplasmic aggregate formation are likely to account for the similar relocalization phenotypes of A3Bctd and A3A despite BORF2 interacting more weakly with the latter protein. In comparison, for KSHV ORF61, swapping L1 of A3Bctd with L1 of A3Gctd slightly decreases relocalization, whereas the remaining L1 swaps have no significant effect (*Figure 4B*, representative images; *Figure 4C–D*, quantification).

## EBV BORF2 interacts with A3Bctd loop 1 to inhibit DNA deaminase activity

In addition to causing protein relocalization, direct binding by EBV BORF2 potently inhibits A3B catalytic activity in vitro (*Cheng et al., 2019b*; *Shaban et al., 2022*). This supports a model in which EBV BORF2 simultaneously inhibits A3B activity and moves the complex away from viral replication centers to protect EBV genomes from deamination during lytic reactivation. Given results here showing that EBV BORF2 and KSHV ORF61 bind to both A3B and A3A, we next asked whether direct enzymatic inhibition is a conserved feature of this pathogen-host interaction.

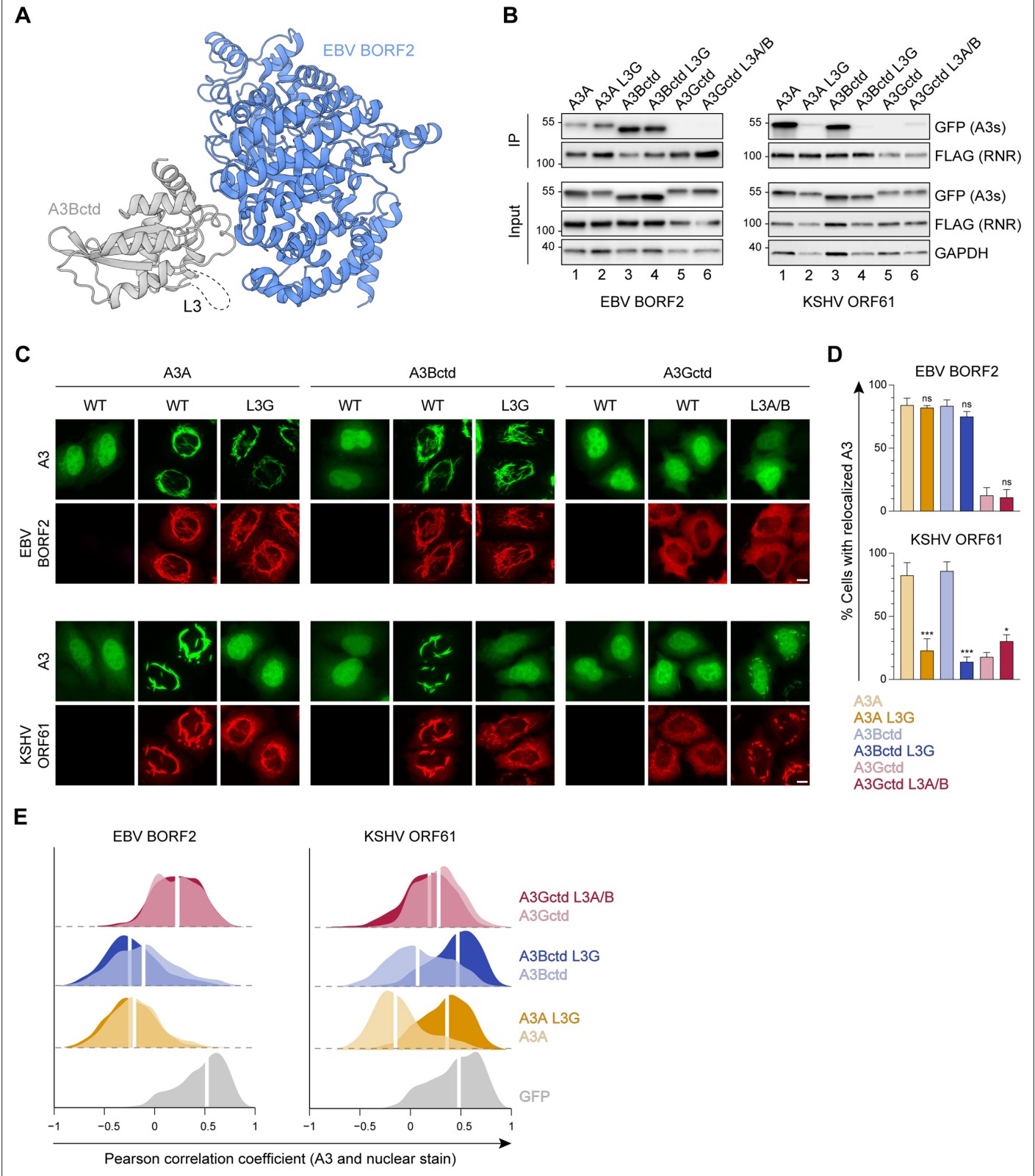

**Figure 3.** Role of A3B and A3A loop 3 in binding to gamma-herpesvirus ribonucleotide reductases (RNRs). (**A**) Ribbon schematic of the Epstein-Barr virus (EBV) BORF2-A3Bctd complex (pdb 7rw6, chains A and B) with BORF2 in blue and A3Bctd in gray. Gray dashed line represents the L3 region not visible by cryo-EM. (**B**) Co-immunoprecipitation (Co-IP) of A3 L3 chimeras with EBV BORF2 and Kaposi's sarcoma-associated herpesvirus (KSHV) ORF61. FLAG-tagged RNR subunits were co-expressed with the indicated A3-EGFP constructs in 293T cells, affinity purified, and analyzed by immunoblotting

*Figure 3 continued on next page*

*Figure 3 continued*

to detect co-purifying A3 proteins. (**C**) Representative IF microscopy images of HeLa cells co-transfected with FLAG-tagged EBV BORF2 or KSHV ORF61 (red) and the indicated A3-EGFP constructs (green). Scale = 10 μm. (**D–E**) Quantification of A3-EGFP relocalization in HeLa cells expressing EBV BORF2 or KSHV ORF61. Panel D shows the percentage of cells exhibiting relocalized A3-EGFP (mean ± standard deviation, n≥100 cells per condition). Statistical analyses were performed using unpaired t-tests to determine significant changes in the relocalization of L3 swapped A3s (darker shade) relative to wild-type (WT) (lighter shade; n=3 independent experimental replicates; ns, not significant p>0.5; *p≤0.5; **p≤0.01; ***p≤0.001). Panel E shows Pearson correlation coefficient (PCC) values for A3-EGFP (or EGFP alone as a control) and Hoechst (nuclear stain). Density curves show the PCC value distribution in each condition, and the white vertical lines indicate the median PCC value (n≥100 cells per condition).

The online version of this article includes the following source data for figure 3:

**Source data 1.** File contains original immunoblots for *Figure 3B*.

HeLa T-REx cells stably expressing doxycycline-inducible A3B-EGFP or A3A-EGFP were transfected with an empty vector control or FLAG-tagged EBV BORF2 and KSHV ORF61. After 24 hr of doxycycline induction, whole-cell extracts were incubated with a fluorescently labeled ssDNA oligonucleotide substrate containing a single A3B/A deamination motif (5′-T$\underline{C}$). A3B/A-mediated deamination of cytosine-to-uracil followed by uracil excision by UDG and abasic site cleavage by sodium hydroxide (NaOH) yields a shorter product that can be detected by SDS-PAGE and fluorescence scanning (see Materials and methods). As expected, EBV BORF2 strongly inhibits A3B deaminase activity in cell lysates as indicated by minimal product accumulation (*Figure 5A*, left panel, compare lanes 1 and 2). KSHV ORF61 also inhibits A3B-catalyzed deamination when compared to a no-RNR control, albeit to a considerably lesser extent compared to EBV BORF2 (*Figure 5A*, left panel, compare lanes 1 and 3).

Surprisingly, neither EBV BORF2 nor KSHV ORF61 substantially inhibits the deaminase activity of A3A (*Figure 5A*, right panel). This stark difference is supported by our cryo-EM structure showing extensive contacts between BORF2 and the L1 and L7 regions of A3Bctd, which results in a high-affinity interaction that blocks the A3B active site from binding to ssDNA substrates and catalyzing deamination (*Shaban et al., 2022*). Specifically, A3B L1 residues 206-PLV-208 contribute to the formation of a hydrophobic pocket in which EBV BORF2 residue Y481 is inserted (depicted in *Figure 5B*). In contrast, this 3-residue motif is absent in A3A L1 (*Figure 2A*). We therefore hypothesized that inhibition of deaminase activity by EBV BORF2 requires not only binding to the catalytic domain through L7, but also a significant interaction with L1.

To further test this idea, we first generated mutants of EBV BORF2 Y481 and examined their ability to inhibit A3B-catalyzed deamination using the HeLa T-REx A3B-EGFP system described above. Mutating EBV BORF2 Y481 to an alanine or a proline effectively prevents inhibition of A3B-catalyzed deamination, likely by weakening the interaction with the L1 region (*Figure 5C*). Next, we compared the ability of EBV BORF2 to inhibit the deaminase activity of WT versus L1 swapped A3B and A3A in HeLa cells transfected with FLAG-tagged BORF2 and EGFP-tagged A3s. In agreement with results from *Figure 5A*, WT A3B deaminase activity is potently inhibited by EBV BORF2 (*Figure 5D*, left, lanes 1 and 2). In contrast, when L1 of A3B is replaced with L1 of A3A, EBV BORF2 completely loses its capacity to inhibit ssDNA deamination (*Figure 5D*, left, lanes 3 and 4). In further agreement with data in *Figure 5A*, WT A3A deaminase activity is not inhibited by EBV BORF2 (*Figure 5D*, right, lanes 1 and 2). Unfortunately, the chimeric A3A L1B construct exhibits very low levels of ssDNA deaminase activity, and it is therefore not possible to determine the susceptibility to inhibition by EBV BORF2 (*Figure 5D*, right, lanes 3 and 4). Nevertheless, these results combine to indicate that the EBV BORF2 interaction with A3B L1 is a requirement for inhibiting deamination and also help explain why A3A activity is not inhibited similarly.

A structural overlay of the cryo-EM structure of the EBV BORF2-A3B complex and the RoseTTAFold-predicted model of KSHV ORF61 indicates that a residue analogous to the interlocking Y481 of BORF2 is lacking in ORF61 (*Figure 5E*). Instead, KSHV ORF61 has a proline (P488) immediately downstream of shared proline and threonine residues. This difference suggests that L1 may be allowed to remain flexible and engage ssDNA substrates when A3B is bound to KSHV ORF61. Alternatively—and perhaps more likely—a lack of interaction with L1 may result in RNR-A3B complexes with lower binding affinity (i.e., lower association rate and/or higher dissociation rate), thereby rendering KSHV ORF61 unable to inhibit A3B-catalyzed deamination to the same extent as EBV BORF2.

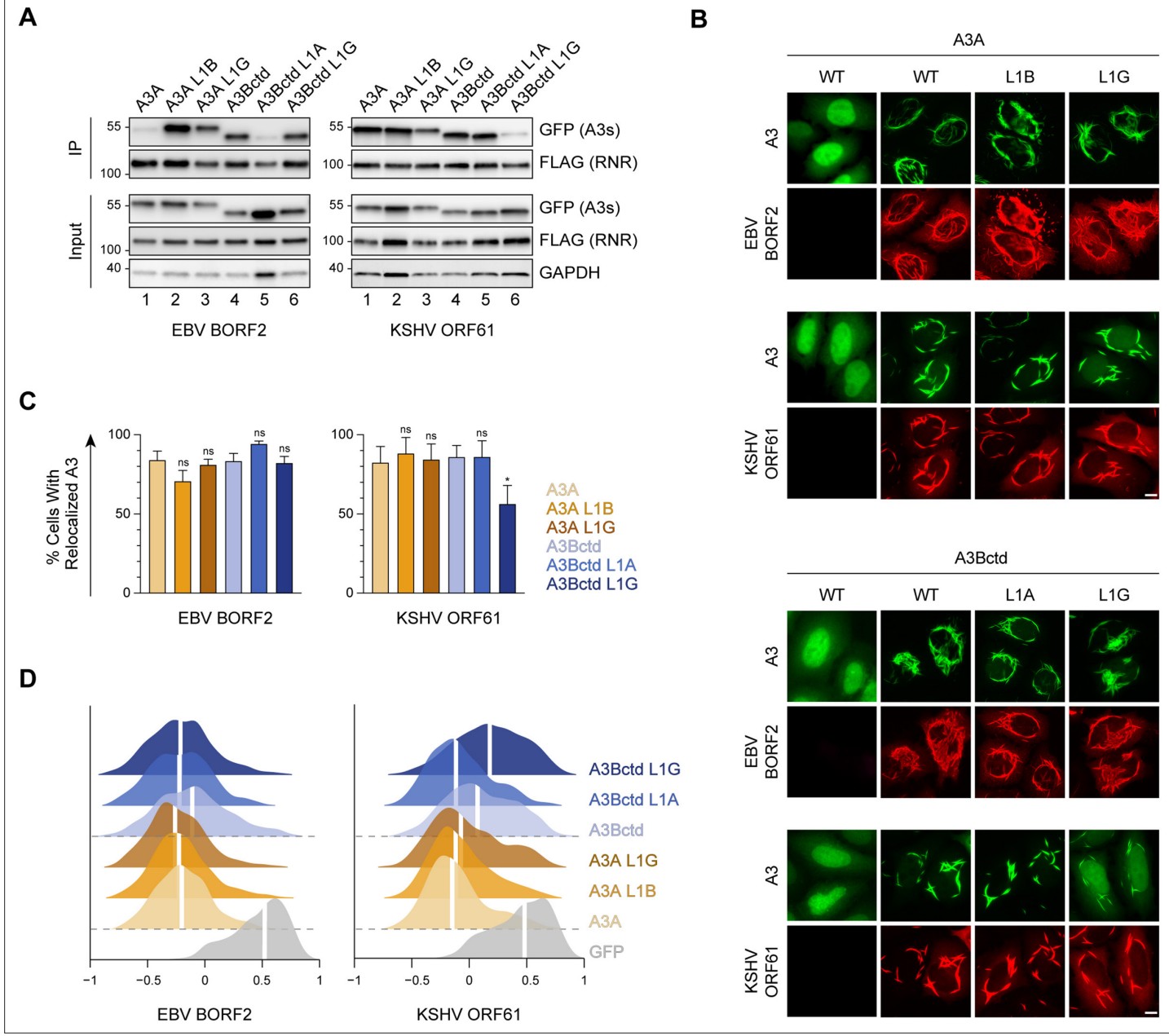

**Figure 4.** Role of A3B and A3A loop 1 in binding to gamma-herpesvirus ribonucleotide reductases (RNRs). (**A**) Co-immunoprecipitation (Co-IP) of A3 L1 chimeras with Epstein-Barr virus (EBV) BORF2 and Kaposi's sarcoma-associated herpesvirus (KSHV) ORF61. FLAG-tagged RNR subunits were co-expressed with the indicated A3-EGFP constructs in 293T cells, affinity purified, and analyzed by immunoblotting to detect co-purifying A3 proteins. (**B**) Representative IF microscopy images of HeLa cells co-transfected with FLAG-tagged EBV BORF2 or KSHV ORF61 (red) and the indicated A3-EGFP constructs (green). Scale = 10 µm. (**C–D**) Quantification of A3-EGFP relocalization in HeLa cells expressing EBV BORF2 or KSHV ORF61. Panel C shows the percentage of cells exhibiting relocalized A3-EGFP (mean ± standard deviation, n≥100 cells per condition). Statistical analyses were performed using unpaired t-tests to determine significant changes in the relocalization of L1 swapped A3s (darker shade) relative to wild-type (WT) (lighter shade; n=3 independent experimental replicates; ns, not significant p>0.5; *p≤0.5; **p≤0.01; ***p≤0.001). Panel D shows Pearson correlation coefficient (PCC) values for A3-EGFP (or EGFP alone as a control) and Hoechst (nuclear stain). Density curves show the PCC value distribution in each condition, and the white vertical lines indicate the median PCC value (n≥100 cells per condition).

The online version of this article includes the following source data for figure 4:

**Source data 1.** File contains original immunoblots for *Figure 4A*.

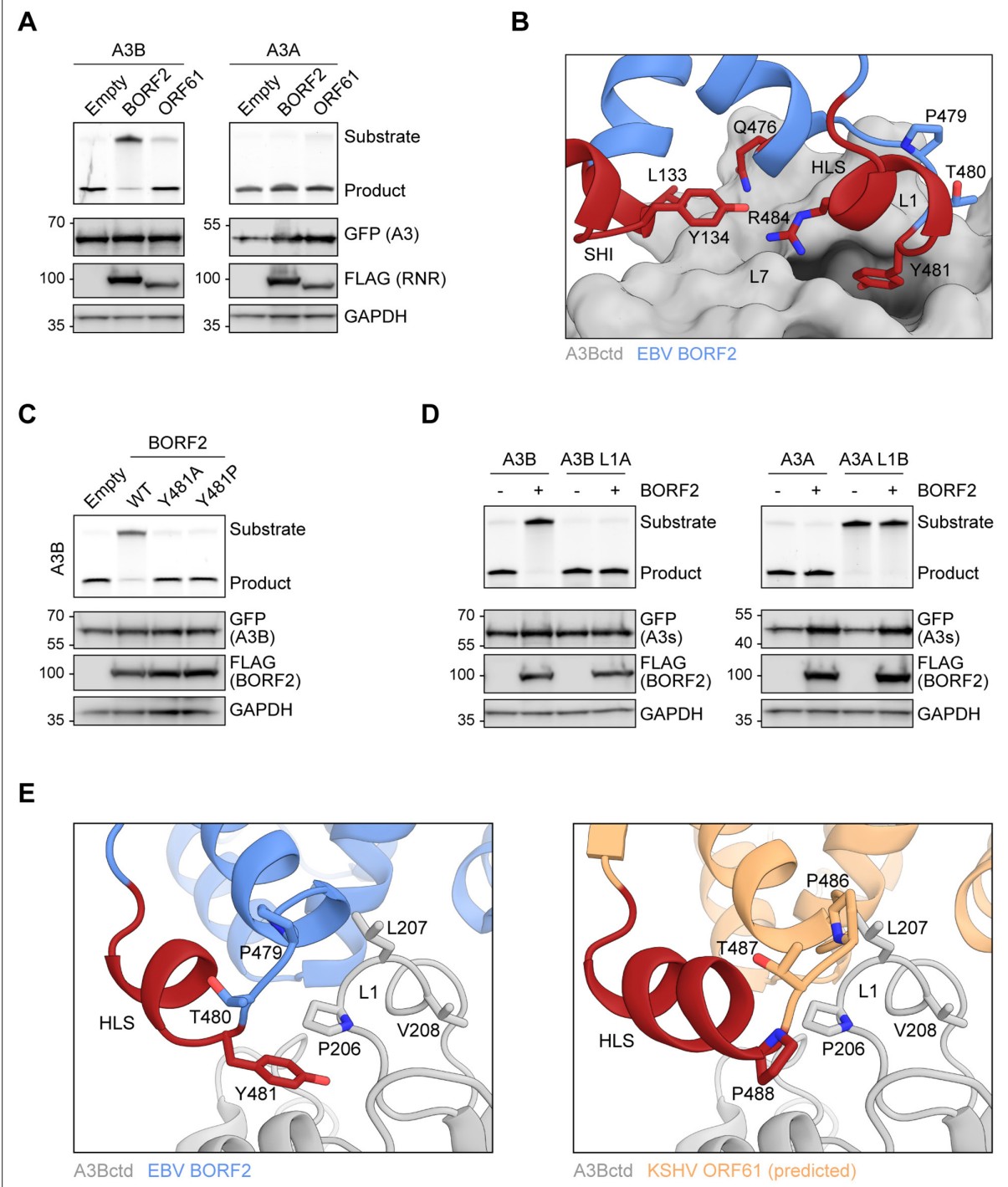

**Figure 5.** Epstein-Barr virus (EBV) BORF2 interacts with A3Bctd loop 1 to inhibit DNA deaminase activity. (**A**) Upper panels show single-stranded DNA (ssDNA) deaminase activity of extracts from HeLa T-REx cells transfected with the indicated FLAG-tagged ribonucleotide reductase (RNR) subunits and incubated with doxycycline to induce the expression of A3B-EGFP or A3A-EGFP. Lower panels show protein expression levels in the same extracts by immunoblotting. (**B**) Zoom-in of the EBV BORF2-A3Bctd interface (pdb 7rw6, chains A and B) showing a network of interactions between BORF2 (blue, ribbon representation) and A3Bctd L7 and L1 regions (gray, surface-filled representation). BORF2 short helix insertion (SHI), helical loop structure (HLS), and Q476 are shown in red. (**C**) Upper panels show ssDNA deaminase activity of extracts from HeLa T-REx cells transfected the indicated FLAG-tagged EBV BORF2 constructs and incubated with doxycycline to induce the expression of A3B-EGFP. Lower panels show protein expression levels in the same extracts by immunoblotting. (**D**) Upper panels show ssDNA deaminase activity of extracts from HeLa cells co-transfected with FLAG-tagged EBV BORF2 and the indicated A3-EGFP constructs. Lower panels show protein expression levels in the same extracts by immunoblotting. (**E**) Ribbon schematic of the EBV BORF2-A3Bctd interface (pdb 7rw6, chains A and B) highlighting BORF2 tyrosine 481 and A3Bctd L1. BORF2 is shown in blue, the HLS region

*Figure 5 continued on next page*

Figure 5 continued

in red, and A3Bctd in gray. The right schematic shows the same view with the predicted structure of Kaposi's sarcoma-associated herpesvirus (KSHV) ORF61 (orange) overlayed and BORF2 residues hidden to allow better visualization (full model in *Figure 1C*).

The online version of this article includes the following source data for figure 5:

**Source data 1.** File contains original deaminase activity assay gels and immunoblots for *Figure 5A*.

**Source data 2.** File contains original deaminase activity assay gels and immunoblots for *Figure 5C*.

**Source data 3.** File contains original deaminase activity assay gels and immunoblots for *Figure 5D*.

## The genesis of *A3B* may have shaped gamma-herpesvirus RNR evolution in primates

*A3* genes are unique to placental mammals and the evolutionary history of the *A3* locus is marked by frequent expansions and contractions resulting from independent events of gene duplication, loss, and fusion (*Münk et al., 2012*; *LaRue et al., 2008*; *Ito et al., 2020*; *Conticello, 2008*). For instance, in the primate lineage leading to humans the *A3* locus evolved from an ancestral 3 deaminase domain cluster to the present-day 11 domain (7 gene) cluster. Interestingly, phylogenetic and comparative genomics studies indicate that *A3B* may be the youngest *A3* family member in primates (*LaRue et al., 2008*; *Uriu et al., 2021b*). Specifically, *A3B* was likely generated by the duplication of ancestral deaminase domains in the common ancestor of Catarrhini primates (i.e., hominoids and Old World monkeys) and is consequently absent in Platyrrhini primates (i.e., New World monkeys) (depicted in *Figure 6A*). We therefore hypothesized that the function of an ancestral gamma-herpesvirus RNR as an innate immune antagonist may have been selected by the birth of *A3B* during this critical period of primate evolution (i.e., 43–29 million years ago, after the split of Catarrhini and Platyrrhini primates).

To test this idea, we cloned the RNR large subunit genes from representative gamma-herpesviruses that infect primates with A3B (EBV and KSHV, which infect humans, and McHV-4 and McHV-5, which infect rhesus macaques) and related viruses that infect primates without A3B (CalHV-3 and SaHV-2, which infect New World marmosets and squirrel monkeys, respectively) (*Figure 6B*). Of note, these RNRs represent gamma-herpesviruses from two evolutionarily diverged genera—*Lymphocryptovirus* and *Rhadinovirus* (*Figure 6B*). We then examined the capacity of these viral RNRs to bind to human A3Bctd and A3A in co-IP experiments (*Figure 6C*). In support of our hypothesis, only the RNRs from viruses that infect A3B-encoding primates (humans and rhesus macaques) pull-down human A3Bctd and A3A (*Figure 6C*, lanes 1–4, left and right panels). In contrast, the RNRs from viruses that infect primates without A3B (marmosets and squirrel monkeys) are unable to pull down either enzyme (*Figure 6C*, lanes 5–6, left and right panels).

We next asked if these viral RNRs can relocalize human A3B and A3A. HeLa T-REx cells were transfected with FLAG-tagged viral RNRs and incubated with doxycycline to induce A3B-EGFP or A3A-EGFP expression (*Figure 6D*). In agreement with our co-IP results, only the RNRs from viruses that infect A3B-encoding primates are able to relocalize human A3B and A3A from the nucleus to the cytoplasmic compartment. Together, these results suggest that the ability to bind and relocalize human A3 enzymes is conserved among RNRs from gamma-herpesviruses that infect Catarrhini primates.

We additionally used HeLa T-Rex cells stably expressing doxycycline-inducible human A3B-EGFP or human A3A-EGFP to ask whether these viral RNRs are capable of inhibiting the ssDNA deaminase activity of human A3B and A3A (*Figure 6E*). Consistent with our previous results, EBV BORF2 potently inhibits A3B (*Figure 6E*, top panel, lane 2). A similarly strong inhibition is seen with the RNR of rhesus macaque *Lymphocryptovirus* McHV-4 (*Figure 6E*, top panel, lane 3). The RNRs from KSHV and rhesus macaque *Rhadinovirus* McHV-5 did not overtly affect the formation of a product band but appeared to slow down A3B-catalyzed deamination, as suggested by a stronger substrate band compared to the no-RNR control (*Figure 6E*, top panel, lanes 5 and 6). Most importantly, the RNRs from marmoset *Lymphocryptovirus* CalHV-3 and squirrel monkey *Rhadinovirus* SaHV-2 are both defective in inhibiting A3B activity (*Figure 6E*, top panel, lanes 4 and 7). Additionally, none of the six different viral RNRs inhibits the activity of human A3A (*Figure 6E*, bottom panel). Collectively, our results suggest a possible link between the birth of *A3B* in the common ancestor of Catarrhini primates and the evolution of A3B-counteraction functionalities in an ancestral gamma-herpesvirus RNR large subunit (i.e., direct binding, subcellular relocalization, and enzymatic inhibition).

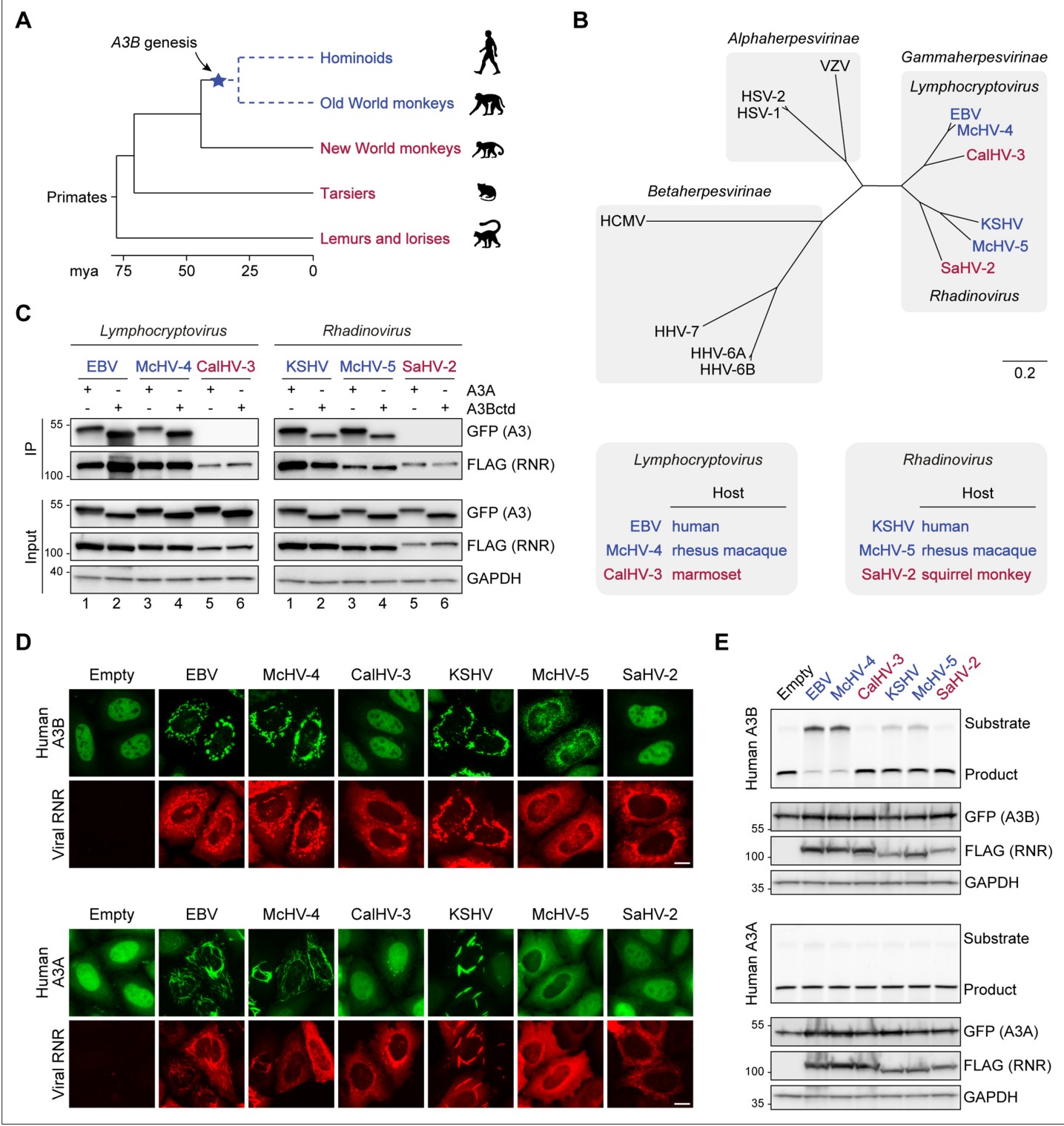

**Figure 6.** The genesis of A3B may have shaped gamma-herpesvirus ribonucleotide reductases (RNRs) evolution. (**A**) Chronogram of primates depicting the generation of *A3B* in the common ancestor of hominoids and Old World monkeys. Primate groups shown in blue are composed of species that encode *A3B* and those shown in red lack *A3B*. Primate phylogeny and divergence intervals were derived from timetree.org (***Kumar et al., 2017***). Mya = million years ago. (**B**) Top: Phylogenetic tree of human herpesvirus and indicated non-human primate herpesvirus RNR large subunits. Shaded boxes indicate alpha-, beta-, and gamma-herpesvirus subfamilies. Genus classification is shown for gamma-herpesviruses. Gamma-herpesviruses that infect host species that encode *A3B* are shown in blue and those that infect host species that lack *A3B* are shown in red. Bottom: Host species infected by the indicated gamma-herpesviruses. Virus-host pairs are colored according to *A3B* status, as described for panel A. (**C**) Co-immunoprecipitation (Co-IP)

*Figure 6 continued on next page*

*Figure 6 continued*

of human A3A and A3Bctd with the RNR large subunits from the indicated viruses. FLAG-tagged RNR subunits were co-expressed with A3A-EGFP or A3Bctd-EGFP in 293T cells, affinity purified, and analyzed by immunoblotting to detect co-purifying A3 proteins. (**D**) Representative IF microscopy images of HeLa T-REx cells transfected with the indicated FLAG-tagged RNR subunits (red) and incubated with doxycycline to induce the expression of human A3B-EGFP or A3A-EGFP (green). Scale = 10 µm. (**E**) Upper panels show single-stranded DNA (ssDNA) deaminase activity of extracts from HeLa T-REx cells transfected with the indicated FLAG-tagged RNR subunits and incubated with doxycycline to induce the expression of A3B-EGFP or A3A-EGFP. Lower panels show protein expression levels in the same extracts by immunoblotting.

The online version of this article includes the following source data for figure 6:

**Source data 1.** File contains original immunoblots for *Figure 6C*.

**Source data 2.** File contains original deaminase activity assay gels and immunoblots for *Figure 6E*.

## A short HLS from EBV BORF2 enables the marmoset CalHV-3 RNR to bind to human A3B and A3A

A3 enzymes are composed of single or double zinc-coordinating (Z) domains that can be grouped into three classes (Z1, Z2, and Z3) based on sequence similarity (*LaRue et al., 2009*). In humans, A3A, A3C, and A3H are single Z domain proteins (Z1, Z2, and Z3, respectively), whereas A3B, A3G, A3D, and A3F harbor two Z domains (Z2-Z1 for A3B and A3G, Z2-Z2 for A3D and A3F). Although New World monkeys lack A3B, the genomes of these species—as well as all primates—encode a single-domain Z1 enzyme homologous to human A3A. Therefore, it is possible that the RNRs of gamma-herpesviruses that infect New World monkeys may have evolved to antagonize the A3A enzymes of their natural host species.

To investigate this possibility, we performed co-IP experiments using FLAG-tagged viral RNRs and host-matched EGFP-tagged A3As (*Figure 7A*). In line with our results above with human A3Bctd and A3A (*Figure 6*), all four RNRs from viruses that infect Catarrhini primates bind to the A3A enzymes of their respective host species (*Figure 7A*, lanes 1, 2, 4, and 5). In contrast, the RNRs from viruses infecting New World monkeys are unable to bind to their hosts' A3As (*Figure 7A*, lanes 3 and 6). We next asked if these RNRs can relocalize the A3A enzymes encoded by their respective host species (*Figure 7B*). In agreement with the co-IP results, the RNRs from EBV/KSHV and McHV-4/McHV-5 relocalize human and rhesus A3A, respectively (*Figure 7B*, left and center). In contrast, the RNRs from CalHV-3 and SaHV-2 do not alter the localization of marmoset and squirrel monkey A3A, respectively (*Figure 7B*, right). These results further support the idea that A3 antagonism by gamma-herpesvirus RNRs is likely to be limited to viruses that infect primates with *A3B* and, additionally, suggest that binding to A3A may be a consequence of similarity to A3Bctd (identical loop 3 and loop 7 regions and >90% overall identity).

Our recent cryo-EM structure of the EBV BORF2-A3Bctd complex revealed that loops 1 and 7 of A3B are sequestered by multiple interactions with BORF2 including contacts with a short HLS involving the critical BORF2 residues Y481 and R484 (*Shaban et al., 2022*; region depicted in red in *Figure 7C* and alternative poses in *Figure 1* and *Figure 5*). Amino acid sequence alignments of the RNRs from human EBV, rhesus macaque McHV-4, and marmoset CalHV-3, which are all members of the *Lymphocryptovirus* genus, show that the former two RNRs share identical amino acid sequences in the HLS region, whereas the latter protein has multiple differences (HLS region alignment in *Figure 7C*). Given the overall similarity between the three-dimensional architecture of EBV BORF2 and the predicted structure of CalHV-3 RNR (*Figure 7—figure supplement 1*), we next asked whether replacing the HLS region of the CalHV-3 RNR with the corresponding motif from EBV BORF2 may be sufficient to enable binding to A3B/A enzymes in co-IP experiments. Interestingly, although the chimeric CalHV-3 RNR construct is still unable to bind marmoset A3A (*Figure 7D*, lane 2), this protein is now able to efficiently bind to both human A3Bctd and A3A (*Figure 7D*, lanes 3 and 4). These results indicate that the HLS region of EBV BORF2 is sufficient to endow a normally non-binding *Lymphocryptovirus* RNR with a capacity to engage both human A3Bctd and A3A, likely by enabling a direct physical interaction with the identical loop 7 region of these enzymes (L7 region alignment in *Figure 7E*).

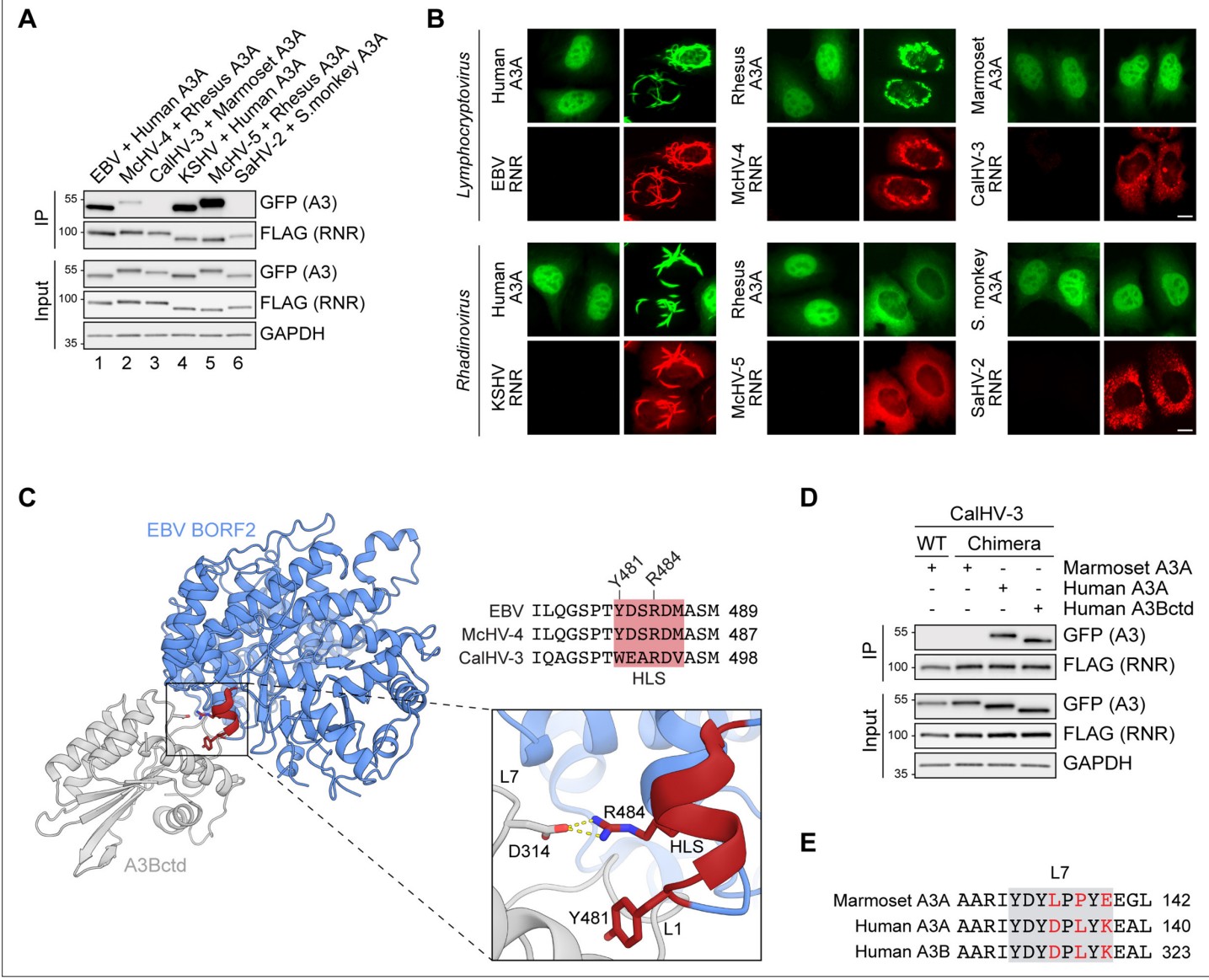

**Figure 7.** A short helical loop structure from Epstein-Barr virus (EBV) BORF2 enables the marmoset CalHV-3 ribonucleotide reductase (RNR) to bind to human A3B and A3A. (**A**) Co-immunoprecipitation (Co-IP) of human, rhesus macaque, marmoset, and squirrel monkey A3As with the indicated gamma-herpesvirus RNR subunits. FLAG-tagged RNR subunits were co-expressed with the indicated A3-EGFP constructs in 293T cells, affinity purified, and analyzed by immunoblotting to detect co-purifying A3 proteins. (**B**) Representative IF microscopy images of HeLa cells co-transfected with the indicated FLAG-tagged viral RNRs (red) and A3A-EGFP constructs (green). Scale = 10 μm. (**C**) Ribbon schematic of the EBV BORF2-A3Bctd complex (pdb 7rw6, chains A and B) with BORF2 in blue and A3Bctd in gray. The helical loop structure (HLS) of EBV BORF2 is depicted in red. Bottom-right: zoom-in of the interacting surfaces highlighting BORF2 Y481 and R484 interactions with A3B loops 1 and 7, respectively. Top-right: amino acid alignment of the HLS regions of the EBV, McHV-4, and CalHV-3 RNRs (red boxes) and adjacent residues. (**D**) Co-IP of marmoset A3A, human A3A, and human A3Bctd with wild-type (WT) or chimeric CalHV-3 RNR containing the HLS region of EBV BORF2. FLAG-tagged RNR subunits were co-expressed with the indicated A3-EGFP constructs in 293T cells, affinity purified, and analyzed by immunoblotting to detect co-purifying A3 proteins. (**E**) Amino acid alignment of the loop 7 regions of marmoset A3A, human A3A, and human A3B (gray boxes) with non-identical loop 7 residues highlighted in red.

The online version of this article includes the following source data and figure supplement(s) for figure 7:

**Source data 1.** File contains original immunoblots for *Figure 7A*.

**Source data 2.** File contains original deaminase activity assay gels and immunoblots for *Figure 7D*.

**Figure supplement 1.** Ribbon schematics of the Epstein-Barr virus (EBV) BORF2 cryo-EM structure (pdb 7rw6, chain A; blue) and the RoseTTAFold-predicted CalHV-3 RNR structure (green).

## Ancestral A3B is active and antagonized by present-day RNRs from gamma-herpesviruses that infect *A3B*-encoding primates

To gain deeper insights into the interaction between A3B and gamma-herpesvirus RNRs, we generated a phylogenetic tree of all publicly available primate A3B sequences and used the GRASP-suite (*Foley et al., 2022*) to reconstruct the amino acid sequence of ancestral A3B (*Figure 8A* and *Figure 8—figure supplement 1*). We then examined an expanded panel of primate gamma-herpesvirus RNRs (i.e., 11 different viral proteins) for a capacity to bind human and ancestral A3Bctd in co-IP experiments (*Figure 8B*). In agreement with results above, only the RNRs from gamma-herpesviruses that infect primates with A3B are capable of binding to human A3Bctd (*Figure 8B*, left). Similarly, nearly all RNRs from viruses that infect primates with A3B are able to bind the ancestral protein and, importantly, all viruses that infect primates which naturally lack the *A3B* gene are unable to bind the ancestral enzyme (*Figure 8B*, right).

To further explore the properties of the ancestral A3B protein, we performed a comprehensive series of subcellular localization experiments and ssDNA deaminase activity assays using this expanded panel of primate gamma-herpesvirus RNRs (*Figure 8C–D* and *Figure 8—figure supplement 2A*). As above, human A3B is completely relocalized from the nucleus to the cytoplasmic compartment by the RNRs of viruses that infect humans and other primates with an *A3B* gene (*Figure 8C*, top; *Figure 8—figure supplement 2A*). Similar results were obtained with ancestral A3B, although only partial relocalization is observed with the KSHV ORF61 (*Figure 8C*, bottom; *Figure 8—figure supplement 2A*). KSHV ORF61 also exhibit a decreased ability to inhibit the deaminase activity of ancestral A3B compared to human A3B (*Figure 8D*, compare lane 5 from the top and bottom panels). These relocalization and ssDNA deaminase activity results correlate with the co-IP observations in *Figure 8B*, where the RNRs from the two human gamma-herpesviruses showed decreased binding to ancestral A3Bctd compared to the human enzyme. Such differences in binding to human versus ancestral A3B may be an indication of adaptive changes in present-day viral RNRs driven by host-pathogen conflicts.

A phylogenetic tree based on the amino acid sequences of available gamma-herpesvirus RNRs indicates that the sequences from human, rhesus macaque, and marmoset *Lymphocryptoviruses* cluster together and the sequences from the different *Rhadinoviruses* cluster according to viral lineage (*Figure 8E*). Interestingly, despite its similarity to KSHV ORF61, the RNR from ColHV-1—a novel KSHV-related virus identified in African *Colobinae* monkeys (*Dhingra et al., 2019*)—differs from the RNRs of all other viruses that infect Catarrhini primates tested here in that it shows no ability to bind, relocalize, or inhibit present-day human or ancestral A3B (*Figure 8B*, *Figure 8—figure supplement 2A*, and *Figure 8D*) or interact with its host's A3A (*Figure 8—figure supplement 3A*). This apparent incongruity may be explained by the fact that African *Colobinae* monkeys appear to have lost the *A3B* gene following the *Colobinae* subfamily split into the African and Asian tribes 10–14 million years ago (*Figure 8—figure supplement 3B–C*), which presumably relieved the pressure for ColHV-1 to maintain an A3B-neutralization activity.

Differential binding of viral RNRs to present-day and ancestral A3B is likely a reflection of past—and possibly still ongoing—genetic conflicts. To further explore this idea, we analyzed a dataset of primate *A3B* DNA sequences for evidence of codon sites under diversifying/positive selection using the mixed effects model of evolution algorithm (MEME) (*Murrell et al., 2012*). We detected evidence for diversifying selection at multiple sites, including clusters of positively selected residues in loops 1 and 3 (*Figure 8F*, top; *Figure 8—figure supplement 4A–B*). This suggests that the amino acid composition of these regions may have been influenced by recurrent diversifying selection, possibly as a mean to escape binding by gamma-herpesvirus RNRs, which in turn would select for virus adaptation to restore binding (depicted in *Figure 8F*, bottom). In line with this interpretation and together with results in *Figure 3* and *Figure 4*, it is possible that differences in positively selected residues in loops 1 and 3 may respectively account for the apparent increased ability of EBV BORF2 and KSHV ORF61 to antagonize present-day human A3B compared to ancestral A3B.

## Discussion

Viruses use a plethora of mechanisms to avoid and antagonize host immune responses. We previously reported an unexpected role for herpesvirus RNRs in counteracting innate immunity by revealing that EBV repurposes its RNR subunit BORF2 to neutralize the antiviral activity of cellular A3B (*Cheng*

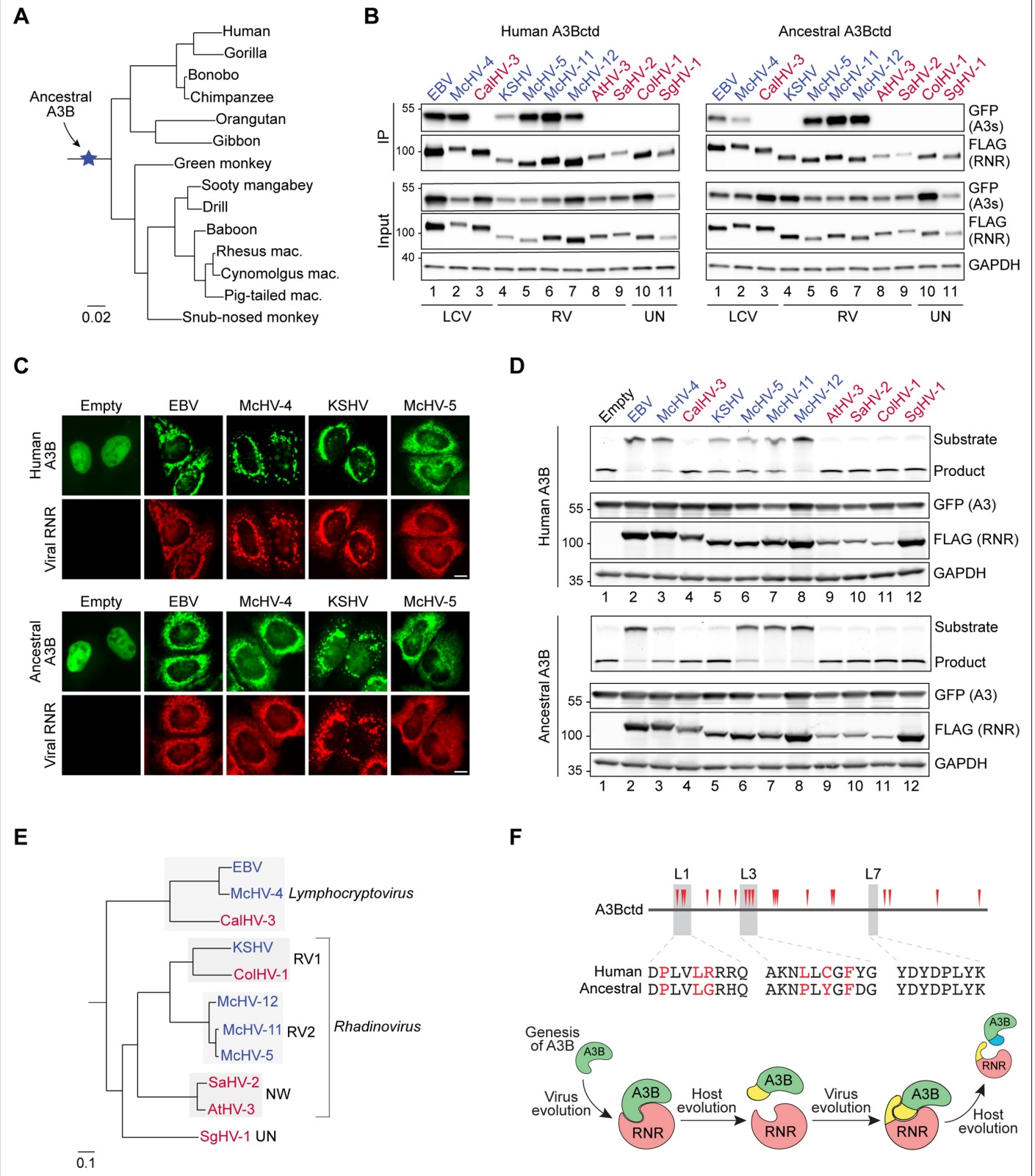

**Figure 8.** Ancestral A3B is active and antagonized by the ribonucleotide reductases (RNRs) from gamma-herpesviruses that infect *A3B*-encoding primates. (**A**) Phylogenetic tree of primate A3B sequences used in this study to reconstruct ancestral A3B (amino acid sequence alignment in *Figure 8—figure supplement 1*). (**B**) Co-immunoprecipitation (Co-IP) of human and ancestral A3Bctd with the indicated gamma-herpesvirus RNR subunits. FLAG-tagged RNR subunits were co-expressed with the indicated A3-EGFP constructs in 293T cells, affinity purified, and analyzed by immunoblotting

*Figure 8 continued*

to detect co-purifying A3 proteins. RNRs from viruses that infect host species that encode *A3B* are shown in blue, and RNRs from viruses that infect host species that lack *A3B* are shown in red. LCV = *Lymphocryptovirus*; RV = *Rhadinovirus*; UN = unclassified gamma-herpesvirus (no official genus classification from the International Committee on Taxonomy of Viruses; **Lefkowitz et al., 2018**). (**C**) Representative IF microscopy images of HeLa cells co-transfected with human or ancestral A3B-EGFP (green) and the indicated FLAG-tagged viral RNRs (red). Scale = 10 μm. See **Figure 8—figure supplement 2A** for additional RNR data. (**D**) Upper panels show single-stranded DNA (ssDNA) deaminase activity of extracts from HeLa cells co-transfected with human or ancestral A3B-EGFP and the indicated FLAG-tagged viral RNRs. Lower panels show protein expression levels in the same extracts by immunoblotting. (**E**) Phylogenetic tree of primate gamma-herpesviruses RNR large subunits. *Rhadinovirus* RV1, RV2, and NW refer to primate *Rhadinovirus* lineages (**Greensill et al., 2000**; **Strand et al., 2000**; **Lacoste et al., 2001**; **Schultz et al., 2000**; **Whitby et al., 2003**) (UN, unclassified). (**F**) Top: positively selected sites in A3Bctd identified using mixed effects model of evolution algorithm (MEME) (**Murrell et al., 2012**) with p<0.1 (red triangles in schematic and red residues in loop region alignments; see **Figure 8—figure supplement 4** for details). Bottom: schematic of the proposed genetic conflict between primate gamma-herpesvirus RNRs and cellular A3B.

The online version of this article includes the following source data and figure supplement(s) for figure 8:

**Source data 1.** File contains original immunoblots for *Figure 8B*.

**Source data 2.** File contains original deaminase activity assay gels and immunoblots for *Figure 8D*.

**Figure supplement 1.** Amino acid sequence alignment of the indicated primate A3Bs.

**Figure supplement 2.** Relocalization of additional viral RNRs by human and ancestral A3B and lack of human A3A inhibition by Catarrhini gamma-herpesvirus RNRs.

**Figure supplement 2—source data 1.** File contains original deaminase activity assay gels and immunoblots for *Figure 8—figure supplement 2*, panel B.

**Figure supplement 3.** Lack of A3-binding by ColHV-1 RNR may be explained by the apparent loss of *A3B* in African colobus monkeys.

**Figure supplement 3—source data 1.** File contains original immunoblots for *Figure 8—figure supplement 3*, panel A.

**Figure supplement 4.** Diversifying positive selection in primate A3B.

*et al., 2019b*). The importance of this A3B counteraction mechanism is evidenced by BORF2-null EBV exhibiting lower infectivity and an accumulation of A3B signature C/G-to-T/A mutations (**Cheng et al., 2019b**). Subsequent studies revealed that the RNR subunits from related human herpesviruses also bind and relocalize A3B, as well as the highly similar A3A enzyme (**Stewart et al., 2019**; **Cheng et al., 2019a**). Here, we leverage knowledge gained from our recent cryo-EM structure of EBV BORF2 in complex with the catalytic domain of A3B (**Shaban et al., 2022**) to investigate the mechanistic conservation of A3 antagonism by herpesviruses. We find that EBV BORF2 and KSHV ORF61 bind to cellular A3 enzymes via distinct surfaces and show that direct inhibition of ssDNA deaminase activity by EBV BORF2 requires specific interactions with the L1 of A3B in addition to the L7 region. We further find that across different primate gamma-herpesvirus lineages, the existence of an RNR-mediated mechanism to counteract cellular A3 enzymes appears to be confined to viral species that infect A3B-expressing primates. Together, these results suggest that the genesis of *A3B* by duplication of ancestral Z2 and Z1 deaminase domains in the common ancestor of present-day hominoids and Old World monkeys may have selected for an ancestral gamma-herpesviruses with a capacity to antagonize this new restriction factor through its large RNR subunit. To our knowledge, this is the first study to link the birth of an antiviral host factor and the evolution of an equally potent viral escape mechanism.

Our data here not only highlight the overall conservation of A3B antagonism by gamma-herpesvirus large RNR subunits but also reveal new and unexpected molecular differences. In agreement with our recent cryo-EM studies (**Shaban et al., 2022**), we show that A3B L7 is a critical determinant of EBV BORF2 binding, whereas A3B L1 serves to help strengthen this interaction and confer selective binding over A3A (**Figure 2** and **Figure 4**). Surprisingly, these same rules do not apply to KSHV ORF61, which appears to mainly rely on L3 to bind and relocalize cellular A3B/A (**Figure 3**). Together, our results suggest a model where interactions with L7 and/or L3—which are identical in A3B and A3A, but not in any other members of the A3 family—enable gamma-herpesvirus RNRs to bind and relocalize both enzymes, whereas additional interacting regions (i.e., EBV BORF2 HLS and A3B L1) may help confer selectivity and additional neutralization activity (e.g., potent inhibition of A3B ssDNA deaminase activity). It is additionally interesting that this may be a new manifestation of a pathogen-host wobble mechanism (**Richards et al., 2015**) in which ongoing small changes over evolutionary time in the viral RNRs can manifest molecularly in the present-day target protein as distinct binding surfaces (e.g., EBV BORF2 requires A3Bctd L7 and KSHV ORF61 requires L3).

Using a panel of 11 primate gamma-herpesvirus RNRs, we show that, in addition to binding and relocalization, all RNRs derived from viruses that infect hosts that encode A3B show some capacity to inhibit the ssDNA deamination activity of human A3B, but none of these viral proteins could inhibit the deaminase activity of human A3A (*Figure 8* and *Figure 8—figure supplement 2B*). This is consistent with a model in which the antiviral activity of A3B may have forced these viruses to evolve and maintain a potent counteraction mechanism, whereas interaction with A3A may be a consequence of sequence homology to A3B and less likely a true physiologic function. This interpretation is further supported by our observation that the RNRs from viruses that infect *A3B*-null primates are incapable of interacting with the A3A enzymes of their natural host species (*Figure 7*).

The A3B-selective nature of this host-pathogen interaction is orthogonally supported by previous binding studies showing that the EBV BORF2-A3Bctd interaction has a biolayer interferometry dissociation constant of ~1 nM, whereas the interaction with the >90% identical A3A enzyme is over 10-fold weaker (*Shaban et al., 2022*). Our results here reveal that, in addition to providing selective binding to A3B, EBV BORF2 interaction with the L1 region of A3B is required for potently inhibiting DNA deaminase activity (*Figure 5*). It is thus possible that the direct binding of EBV BORF2 to two loop regions near the A3B catalytic site—L1 and L7—helps stabilize the BORF2-A3B complex, resulting in a high affinity interaction and consequently blocking the A3B active site from accessing ssDNA substrates and catalyzing deamination. Importantly, the potential adaptability of this A3B counteraction mechanism in *Lymphocryptovirus*es is highlighted by results showing that grafting the HLS region from EBV BORF2—which contains residues that directly contact A3B L1 and L7—into the RNR from marmoset CalHV-3 enables binding to human A3B (*Figure 7*). In comparison, KSHV ORF61 binding to A3B largely through L3 may result in a lower affinity interaction that more weakly inhibits ssDNA deamination. These inferences support a model in which binding to the A3 active site by viral RNRs is responsible for both protein relocalization and enzymatic inhibition, yet substantial deaminase activity inhibition is only observed above a certain affinity threshold. We therefore postulate that preferential binding to A3B might have been an evolutionary adaptation resulting from selective pressures disproportionally affecting gamma-herpesviruses within the *Lymphocryptovirus* genus, such as EBV and McHV-4. For instance, EBV latency is limited to B lymphocytes (*Young and Rickinson, 2004*), which can express high levels of endogenous A3B (*Koning et al., 2009*; *Refsland et al., 2010*), strongly implying that the neutralization of this enzyme is required to preserve viral genome integrity upon lytic reactivation. On the other hand, the broader tropism of KSHV suggests that lytic replication might not always occur in the presence of A3B (*Blasig et al., 1997*; *Caselli et al., 2005*; *Wu et al., 2006*; *Kim et al., 2003*; *Chakraborty et al., 2012*), which may result in lower exposure to A3B-mediated deamination.

Evolutionary analyses of host-pathogen antagonistic relationships are powerful tools for understanding protein functions. The link between *A3B* generation by duplication of ancestral Z2 and Z1 domains and the evolution of A3-neutralization functionalities in herpesvirus RNRs described here suggests that the double domain architecture of A3B may be a requirement for herpesvirus restriction. Notably, the non-catalytic Z2 domain of A3B has been shown to account for this enzyme's distinct nuclear localization (*Salamango et al., 2018*) and association with multiple RNA binding proteins, in addition to allosterically regulating the catalytic activity of the Z1 domain (*Xiao et al., 2017*). It is therefore plausible that cooperation between Z2 and Z1 domains accounts for A3B antiviral function, possibly by facilitating access to herpesvirus transcription complexes inside the nucleus (e.g., ssDNA exposed in viral R-loop structures; *McCann et al., 2021*). Future studies dissecting the molecular mechanisms that allow A3B to recognize and target foreign nucleic acids such as herpesvirus genomes might reveal insights into the regulation of ssDNA deaminase activity and how the loss of such regulatory mechanisms can at times result in cellular chromosomal DNA mutations and cancer development (reviewed by *Venkatesan et al., 2018*; *Olson et al., 2018*). Future work will likely benefit from interrogating the RNRs and A3 enzymes from a larger and more diverse panel of virus and host species, which is currently prevented by the relatively small number of herpesvirus complete genomes deposited in public databases. Bat species may be particularly interesting given they encode the largest known A3 repertoire in mammals (*Hayward et al., 2018*; *Jebb et al., 2020*) and are also known to host a variety of different herpesviruses (*Wibbelt et al., 2007*; *Watanabe et al., 2010*; *Wu et al., 2012*; *Anthony et al., 2013*; *Sasaki et al., 2014*; *Sano et al., 2015*; *Zheng et al., 2016*; *Shabman et al., 2016*; *Escalera-Zamudio et al., 2016*; *James et al., 2020*; *Letko et al., 2020*).

## Materials and methods

### DNA constructs

The large RNR subunit from EBV (HHV-4), McHV-4, CalHV-3, KSHV (HHV-8), McHV-5, McHV-11, McHV-12, AtHV-3, SaHV-2, ColHV-1, and SgHV-1 were obtained as gBlocks purchased from Integrated DNA Technologies and cloned into pcDNA4/TO (Invitrogen #V102020) with a C-terminal 3×FLAG tag (DYKDDDDK) by restriction digestion and ligation. EBV BORF2$_{Y481A}$, EBV BORF2$_{Y481P}$, CalHV-3 RNR$_{WEARDV490YDSRDM}$ constructs were generated by site-directed mutagenesis using Q5 High-Fidelity DNA Polymerase (NEB #M0491). WT constructs match the following GenBank accessions: EBV BORF2 (V01555.2), McHV-4 BORF2 (AY037858.1), CalHV-3 ORF55 (AF319782.2), KSHV ORF61 (U75698.1), McHV-5 ORF61 (AF083501.3), McHV-11 ORF61 (AY528864.1), McHV-12 ORF61 (KP265674.2), AtHV-3 ORF61 (AF083424.1), SaHV-2 ORF61 (X64346.1), ColHV-1 ORF61 (MH932584.1), and SgHV-1 ORF26 (OK337614.1).

APOBEC3 enzymes were obtained as gBlocks purchased from Integrated DNA Technologies and cloned into pcDNA5/TO (Invitrogen #V103320) with a C-terminal EGFP-tag by restriction digestion and ligation. The loop swap constructs A3B_L1A3A$_{25-30}$, A3Bctd$_{193-382}$_L1A3A$_{25-30}$, A3Bctd$_{193-382}$_L1A3G$_{209-217}$, A3Bctd$_{193-382}$_L3A3G$_{247-254}$, A3Bctd$_{193-382}$_L7A3G$_{317-322}$, A3A_L1A3B$_{205-213}$, A3A_L1A3G$_{209-217}$, A3A_ L3A3G$_{247-254}$, A3A_L7A3G$_{317-322}$, A3Gctd$_{197-384}$_L3A3A$_{60-67}$, and A3Gctd$_{197-384}$_L7A3A$_{132-137}$ were generated by site-directed mutagenesis using Q5 High-Fidelity DNA Polymerase (NEB #M0491). WT constructs match the following GenBank accessions: *Homo sapiens* A3B (NM_004900), *Homo sapiens* A3A (NM_145699), *Homo sapiens* A3G (NM_021822), *Macaca mulatta* A3A (NM_001246231.1), *Macaca mulatta* A3B (NM_001246230.2), *Callithrix jacchus* A3A (NM_001301845.1), *Saimiri boliviensis* A3A (XM_039462978.1), *Colobus angolensis palliatus* A3A (XM_011964212.1).

### Human cell culture

Unless indicated, cell lines were derived from established laboratory collections. All cell cultures were supplemented with 10% fetal bovine serum (Gibco #26140-079), 1× penicillin-streptomycin (Gibco #15140-122), and periodically tested for mycoplasma contamination using MycoAlert PLUS Mycoplasma Detection Kit (Lonza #LT07-710). 293T cells were cultured in high glucose DMEM (HyClone #SH30022.01) and HeLa cells were cultured in RPMI 1640 (Gibco #11875-093).

To generate HeLa T-REx A3A-EGFP and A3B-EGFP cell lines, HeLa cells were transduced with pLENTI6/TR (Invitrogen #V480-20), selected with 5 µg/ml blasticidin (GoldBio #B-800-25), and subcloned by serial dilution to obtain single cell clones. Clonal populations were stably transfected with pcDNA5/TO-A3A-EGFP or pcDNA5/TO-A3B-EGFP, selected with 200 µg/ml hygromycin (GoldBio #H-270-1), and subcloned by serial dilution. Representative clones were used for DNA deaminase activity assays (*Figure 5A*, *Figure 5C*, *Figure 6E*) and IF microscopy experiments (*Figure 6D*).

### Co-IP experiments

Approximately 250,000 293T cells were seeded in six-well plates and transfected the next day with 100 ng pcDNA5/TO-A3-EGFP and 200 ng pcDNA4/TO-RNR-3×FLAG using TransIT-LT1 (Mirus #2304). Cells were harvested ~24 hr after transfection and resuspended in 500 µl of lysis buffer (50 mM Tris-HCl pH 7.4, 150 mM NaCl, 10% glycerol, 0.5% IGEPAL, 100 µg/ml RNaseA, 1 tablet of Sigma-Aldrich cOmplete Protease Inhibitor Cocktail). Resuspensions were vortexed, sonicated on ice-bath for 10 s at the lowest setting using a Branson sonifier, and centrifuged at 15,000 × *g* at 4°C for 15 min. A 50 µl aliquot was removed from each sample for input detection and the remaining clarified whole-cell lysate was incubated with 20 µl packed gel volume of anti-FLAG M2 magnetic beads (Sigma-Aldrich #M8823) at 4°C overnight with gentle rotation. Samples were washed three times with 500 µl of lysis buffer and bound protein was eluted with 50 µl of elution buffer (100 µg/ml FLAG peptide [Sigma-Aldrich #F3290], 50 mM Tris-HCl pH 7.4, 150 mM NaCl). Input and elution samples were analyzed by immunoblot using mouse anti-GFP (1:5000, Clontech #632381) to detect A3-EGFP, rabbit anti-FLAG (1:5,000, Sigma-Aldrich #F7425) to detect RNR-3×FLAG, and rabbit anti-GAPDH (1:5000, Proteintech #60004) to detect GAPDH (loading control). Secondary antibodies used were IRDye 800CW anti-rabbit (LI-COR #926-32211) and HRP-linked anti-mouse (Cell Signaling #7076), both at 1:10,000 dilution.

## Immunofluorescence microscopy experiments

In *Figure 6D*, approximately 5000 engineered HeLa T-REx cells (described above) were seeded into 96-well imaging plates (Corning #3904) and transfected the next day with 30 ng pcDNA4/TO-RNR-3×FLAG. A3B-EGFP or A3A-EGFP expression was induced 6 hr after transfection with 50 ng/ml doxycycline. Approximately 24 hr after induction, cells were fixed in 4% formaldehyde (Thermo Fisher #28906), permeabilized in 0.2% Triton X-100 (Sigma-Aldrich #T8787) for 10 min and incubated in blocking buffer (2.8 mM $KH_2PO_4$, 7.2 mM $K_2HPO_4$, 5% goat serum (Gibco #16210064), 5% glycerol, 1% cold water fish gelatin (Sigma-Aldrich #G7041), 0.04% sodium azide, pH 7.2) for 1 hr. Cells were then incubated with primary antibody mouse anti-FLAG (1:1000, Sigma #F1804) overnight at 4°C with gentle rocking to detect FLAG-tagged viral RNRs. The next day, cells were incubated with secondary antibody anti-mouse Alexa Fluor 594 (1:1000, Invitrogen #A-11005) for 2 hr at room temperature protected from light and nuclei were counterstained with Hoechst 33342 (Thermo Fisher #62249) for 10 min. Representative images were acquired using a Cytation 5 Cell Imaging Multi-Mode Reader (BioTek) at ×20 magnification.

In *Figures 2–4*, approximately 5000 HeLa cells were seeded into 96-well imaging plates (Corning #3904) and transfected the next day with 15 ng pcDNA5/TO-A3-EGFP and 30 ng pcDNA4/TO-RNR-3×FLAG. Cells were fixed approximately 24 hr after transfection. Permeabilization, antibody staining, and nuclei counterstaining with Hoechst 33342 were performed as described above. Plates were imaged using a Cytation 5 Cell Imaging Multi-Mode Reader (BioTek) at ×20 magnification using the automated imaging function to capture 50 non-overlapping images per well. Background subtraction was performed using Fiji (*Schindelin et al., 2012*) and images were analyzed using CellProfiler v4.2 (*Stirling et al., 2021b*). A pipeline was generated to identify individual nuclei, trace cell boundaries, segment the cytoplasmic/nuclear compartments, and take measurements of intensity, texture, and correlation for the Hoechst (nuclear stain), RNR, and A3 channels within each subcellular compartment. At least 100 cells were measured for each well. All measurements were exported to an SQLite database and CellProfiler Analyst v3.0 (*Stirling et al., 2021a*) was used for scoring phenotypes by machine learning in order to quantify the number of cells showing relocalized versus non-relocalized A3 within the population of RNR-expressing cells in each well. The percentages of cells with relocalized A3 in *Figure 2D*, *Figure 3D*, and *Figure 4C* were calculated for n=3 independent experimental replicates (i.e., data generated from the analysis of cells seeded, transfected, and imaged on three different days). The ridgeline plots in *Figure 2E*, *Figure 3E*, and *Figure 4D* were generated in RStudio with the ggplot2 package (*Wickham, 2016*) using the Pearson correlation coefficient measurements for the A3 and Hoechst channels within each segment cell nucleus.

In *Figure 7B* and *Figure 8C*, approximately 5000 HeLa cells were seeded into 96-well imaging plates (Ibidi #89626) and transfected the next day with 15 ng pcDNA5/TO-A3-EGFP and 30 ng pcDNA4/TO-RNR-3×FLAG. Cells were fixed approximately 24 hr after transfection. Permeabilization, antibody staining, and nuclei counterstaining with Hoechst 33342 were performed as described above. Plates were imaged using an Eclipse Ti2-E inverted microscope (Nikon) at ×20 magnification.

## ssDNA deaminase activity assays

In *Figure 5A*, *Figure 5C*, and *Figure 6E*, approximately 250,000 HeLa T-REx cells stably expressing doxycycline-inducible A3B-EGFP or A3A-EGFP (described above) were seeded into six-well plates and transfected the next day with 200 ng pcDNA4/TO-RNR-3×FLAG using TransIT-LT1 (Mirus #2304). A3B-EGFP or A3A-EGFP expression was induced 6 hr after transfection with 50 ng/ml doxycycline. Approximately 24 hr after induction, cells were harvested, resuspended in 100 µl of reaction buffer (25 mM HEPES, 15 mM EDTA, 10% glycerol, 1 tablet of Sigma-Aldrich cOmplete Protease Inhibitor Cocktail), and subjected to freeze-thaw. Whole-cell lysates were then centrifuged at 15,000 × *g* for 15 min and the clarified supernatant was transferred to a new tube. Soluble lysates were incubated with 100 µg/ml RNAse A at room temperature for 1 hr. A 5 µl aliquot was removed from each sample for immunoblots using mouse anti-GFP (1:5000, Clontech #632381) to detect A3-EGFP, rabbit anti-FLAG (1:5000, Sigma-Aldrich #F7425) to detect RNR-3×FLAG, and rabbit anti-GAPDH (1:5000, Proteintech #60004) to detect GAPDH (loading control). Secondary antibodies used were IRDye 800CW anti-rabbit (LI-COR #926-32211) and HRP-linked anti-mouse (Cell Signaling #7076), both at 1:10,000 dilution. Five µl of soluble lysate were incubated at 37°C for 1 hr with 0.7 µM of a fluorescent ssDNA substrate (5′-ATT ATT ATT AT<u>T CA</u>A ATG GAT TTA TTT ATT TAT TTA TTT ATT

T-fluorescein-3') and 2.5 U of UDG (NEB #M0280) in a total reaction volume of 10 μl (diluted in reaction buffer). NaOH was then added to the reaction mix to a final concentration of 100 mM and samples were incubated at 98°C for 10 min. The reaction was then mixed with 11 μl 2× formamide buffer (80% formamide, 1×TBE, bromophenol blue, and xylene cyanol) and reaction products were separated on a 15% TBE-urea PAGE gel. Separated DNA fragments were visualized on a Typhoon FLA 7000 scanner (GE Healthcare) on fluorescence mode.

In *Figure 5D* and *Figure 8D*, approximately 250,000 293T cells were seeded in six-well plates and transfected the next day with 100 ng pcDNA5/TO-A3-EGFP and 200 ng pcDNA4/TO-EBV-BORF2-3×FLAG. Cells were harvested approximately 24 hr after transfection and the DNA deaminase activity assay proceeded as described above.

### Protein structure modeling

Protein structure models were generated with RoseTTAFold (*Baek et al., 2021*) using the ROBETTA server followed by simple all-atom refinement of the top-ranked models using the Relax application in the ROSIE 2 server (*Lyskov et al., 2013*). To model the structure of the ORF61-A3Bctd complex (*Figure 1*), RoseTTAFold-predicted models of KSHV ORF61 and A3Bctd were aligned to the cryo-EM BORF2-A3Bctd complex (pdb: 7RW6) using PyMOL (*DeLano, 2002*) to arrange the starting docking pose and local protein-protein docking was performed using the RosettaDock 4.0 server (*Lyskov et al., 2013*; *Marze et al., 2018*). The 10,000 generated structures were sorted based on CAPRI score (*Lensink et al., 2020*) and subsequently ranked according to their interface energy score (*Figure 1—figure supplement 1B*). All protein structure figures were made using PyMOL (*DeLano, 2002*).

### Herpesvirus RNR phylogenetic trees generation

Amino acid sequences of the large RNR subunits from the following herpesviruses were obtained from the NCBI Protein RefSeq database: HSV-1 (YP_009137114.1), HSV-2 (YP_009137191.1), VZV (NP_040142.1), EBV (YP_401655.1), HCMV (YP_081503.1), HHV-6A (NP_042921.1), HHV-6B (NP_050209.1), HHV-7 (YP_073768.1), KSHV (YP_001129418.1), McHV-4 (YP_067953.1), CalHV-3 (NP_733909.1), McHV-5 (NP_570809.1), McHV-11 (AAT00096.1), McHV-12 (AJE29717.1), AtHV-3 (AAC95585.1), SaHV-2 (NP_040263.1), ColHV-1 (QDQ69272.1), and SgHV-1 (UNP64460.1). To generate the phylogenetic trees in *Figure 6B* and *Figure 8E*, protein sequences were aligned using MUSCLE (*Edgar, 2004*) and maximum likelihood phylogenies were inferred using PhyML 3.0 (*Guindon et al., 2010*) using the Smart Model Selection (SMS) tool (*Lefort et al., 2017*) and 100 bootstraps.

### Ancestral APOBEC3B sequence reconstruction

To generate the phylogenetic tree in *Figure 8A*, primate A3B amino acid sequences were aligned using MUSCLE (*Edgar, 2004*; *Figure 8—figure supplement 1*) and a maximum likelihood phylogeny was inferred using PhyML 3.0 (*Guindon et al., 2010*) using the SMS tool (*Lefort et al., 2017*) and 100 bootstraps. Reconstruction of the ancestral A3B protein sequence was performed with GRASP (*Foley et al., 2022*) using the sequence alignment and phylogenetic tree above.

### Positive selection analysis

Primate *A3B* DNA sequences were aligned using ClustalOmega (*Sievers et al., 2011*) and a maximum likelihood phylogeny was inferred using PhyML 3.0 (*Guindon et al., 2010*) using the SMS tool (*Lefort et al., 2017*) and 100 bootstraps. Codon sites under episodic diversifying selection were identified with the HyPhy 2.5 software package (*Pond and Muse, 2005*) using the MEME (*Murrell et al., 2012*) with 100 bootstraps and p-value threshold of 0.1 (*Figure 8—figure supplement 4A–B*).

## Acknowledgements

We thank members of the Harris lab for thoughtful comments. These studies were supported in part by grants to RSH from the National Institute for Allergy and Infectious Diseases (NIAID) R37-AI064046 and the National Cancer Institute (NCI) P01-CA234228. Salary support for SNM was provided by NIAID F31-AI161910 and subsequently by a grant to the University of Minnesota (SNM and RSH) from the Howard Hughes Medical Institute through the James H Gilliam Fellowships for Advanced Study program. Salary support for JTB was provided by NIAID F32-AI147813. Salary support for NMS and AAA was provided in part by NIAID R56-AI150402 and NIH T32-AI83196, respectively. RSH is

the Ewing Halsell President's Council Distinguished Chair and an Investigator of the Howard Hughes Medical Institute. The authors have no competing interests to declare.

## Additional information

### Funding

| Funder | Grant reference number | Author |
|---|---|---|
| National Institute of Allergy and Infectious Diseases | R37-AI064046 | Reuben S Harris |
| National Cancer Institute | P01-CA234228 | Reuben S Harris |
| National Institute of Allergy and Infectious Diseases | F31-AI161910 | Sofia N Moraes |
| National Institute of Allergy and Infectious Diseases | F32-AI147813 | Jordan T Becker |
| National Institute of Allergy and Infectious Diseases | R56-AI150402 | Nadine M Shaban Ashley A Auerbach |
| Howard Hughes Medical Institute | James H Gilliam Fellowships for Advanced Study program | Sofia N Moraes Reuben S Harris |
| National Institutes of Health | T32-AI83196 | Nadine M Shaban Ashley A Auerbach |

The funders had no role in study design, data collection and interpretation, or the decision to submit the work for publication.

### Author contributions

Sofia N Moraes, Conceptualization, Formal analysis, Funding acquisition, Investigation, Visualization, Methodology, Writing - original draft, Writing - review and editing; Jordan T Becker, Conceptualization, Formal analysis, Supervision, Methodology, Writing - review and editing; Seyed Arad Moghadasi, Conceptualization, Formal analysis, Methodology, Writing - review and editing; Nadine M Shaban, Ashley A Auerbach, Formal analysis, Methodology, Writing - review and editing; Adam Z Cheng, Formal analysis, Supervision, Methodology, Writing - review and editing; Reuben S Harris, Conceptualization, Formal analysis, Supervision, Funding acquisition, Writing - original draft, Project administration, Writing - review and editing

### Author ORCIDs

Sofia N Moraes ![ORCID] http://orcid.org/0000-0003-0838-0047
Jordan T Becker ![ORCID] http://orcid.org/0000-0002-0239-5443
Ashley A Auerbach ![ORCID] http://orcid.org/0000-0001-8505-5905
Reuben S Harris ![ORCID] http://orcid.org/0000-0002-9034-9112

### Decision letter and Author response

Decision letter https://doi.org/10.7554/eLife.83893.sa1
Author response https://doi.org/10.7554/eLife.83893.sa2

## Additional files

### Supplementary files
• MDAR checklist

• Source data 1. Numerical data for the bar plots and ridgeline plots in *Figure 2*, *Figure 3*, and *Figure 4*.

## Data availability

All data generated and analyzed during this study are included in the manuscript and supporting files. Numerical data for the bar plots and ridgeline plots in Figure 2, Figure 3, and Figure 4 are provided in Source data 1. Source Data Files containing uncropped gel images and raw files for immunoblots and deaminase activity assays are provided for Figure 2, Figure 3, Figure 4, Figure 5, Figure 6, Figure 7, Figure 8, Figure 8-figure supplement 2, and Figure 8-figure supplement 3. GenBank codes for the sequences used in this study are listed in the Materials and Methods section.

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
